# Non-microtubule tubulin-based backbone and subordinate components of postsynaptic density lattices

Tatsuo Suzuki[1] , Nobuo Terada[2], Shigeki Higashiyama[3], Kiyokazu Kametani[4], Yoshinori Shirai[1] , Mamoru Honda[5], Tsutomu Kai[5], Weidong Li[6,7], Katsuhiko Tabuchi[1,8]

A purification protocol was developed to identify and analyze the component proteins of a postsynaptic density (PSD) lattice, a core structure of the PSD of excitatory synapses in the central nervous system. "Enriched"- and "lean"-type PSD lattices were purified by synaptic plasma membrane treatment to identify the protein components by comprehensive shotgun mass spectrometry and group them into minimum essential cytoskeleton (MEC) and non-MEC components. Tubulin was found to be a major component of the MEC, with non-microtubule tubulin widely distributed on the purified PSD lattice. The presence of tubulin in and around PSDs was verified by post-embedding immunogold labeling EM of cerebral cortex. Non-MEC proteins included various typical scaffold/adaptor PSD proteins and other class PSD proteins. Thus, this study provides a new PSD lattice model consisting of non-microtubule tubulin-based backbone and various non-MEC proteins. Our findings suggest that tubulin is a key component constructing the backbone and that the associated components are essential for the versatile functions of the PSD.

## Introduction

Structural changes in postsynaptic density (PSD) are important mechanisms for maintaining synaptic plasticity, the basis for memory and learning (Bosch et al, 2014). The molecular mechanism underlying PSD remodeling is not currently known, although the role of actin dynamics in spine morphology is well known (Sekino et al, 2007; Bosch et al, 2014; Spence & Soderling, 2015). A complete understanding of the structure of PSD is indispensable to fully elucidate the molecular mechanisms of spine and PSD dynamics during the expression of synaptic plasticity. In a previous study, we

purified a PSD lattice (PSDL) structure and proposed "PSDL-based dynamic nanocolumn" model for the molecular architecture of PSD. In this model, the scaffold protein model and the PSDL model are combined (Suzuki et al, 2018). However, we were unable to elucidate the molecular components of the PSDL because of the insolubility of the protein components. Therefore, whole components could not be identified by SDS–PAGE and Western blotting.

In this study, we developed a purification protocol that avoided the aggregation or denaturation of the PSDL proteins. This new PSDL preparation method is more physiological and has allowed for the identification and analysis of component proteins by SDS–PAGE and Western blotting. This method was used to purify and analyze "lean-" and "enriched"-type PSDLs, which led to the development of a new PSDL model consisting of a non-microtubule (non-MT) tubulin backbone structure and associated proteins.

## Results

### Purification of PSDL using new method

In the initial purification protocol for the PSDL, an ultracentrifugation step was added before the sucrose density gradient (SDG)-ultracentrifugation step to separate the cytoskeletal and soluble proteins (Fig 1A) (Suzuki et al, 2018). In the new method, we inserted this ultracentrifugation step after the SDG ultracentrifugation (Fig 1B). Additional ultracentrifugation was required to remove non-structural soluble and detergent-solubilized proteins from the PSDL preparation to identify the protein components of the PSDL. As a result, the protein solubility of the PSDL preparation to the MPEX PTS reagent, a protein solubilizing reagent used for mass spectrometry (MS), was greatly improved (Fig S1B). This improvement was also observed in the SDS–PAGE results (see the following

[1]Department of Molecular and Cellular Physiology, Shinshu University Academic Assembly, Institute of Medicine, Shinshu University Academic Assembly, Matsumoto, Japan    [2]Health Science Division, Department of Medical Sciences, Graduate School of Medicine, Science and Technology, Shinshu University, Matsumoto, Nagano, Japan    [3]Department of Cell Growth and Tumor Regulation, Proteo-Science Center, Ehime University, To-on, Ehime, Japan    [4]Department of Veterinary Anatomy, Faculty of Veterinary Medicine, Rakuno Gakuen University, Ebetsu, Japan    [5]Bioscience Group, Center for Precision Medicine Supports, Pharmaceuticals and Life Sciences Division, Shimadzu Techno-Research, INC, Kyoto, Japan    [6]Bio-X Institutes, Key Laboratory for the Genetics of Development and Neuropsychiatric Disorders (Ministry of Education), Shanghai Key Laboratory of Psychotic Disorders, and Brain Science and Technology Research Center, Shanghai Jiao Tong University, Shanghai, China    [7]Institute for Biomedical Sciences, Interdisciplinary Cluster for Cutting Edge Research Shinshu University, Matsumoto, Japan    [8]Department of Biological Sciences for Intractable Neurological Diseases, Institute for Biomedical Sciences, Interdisciplinary Cluster for Cutting Edge Research Shinshu University, Matsumoto, Japan

Correspondence: suzukit@shinshu-u.ac.jp

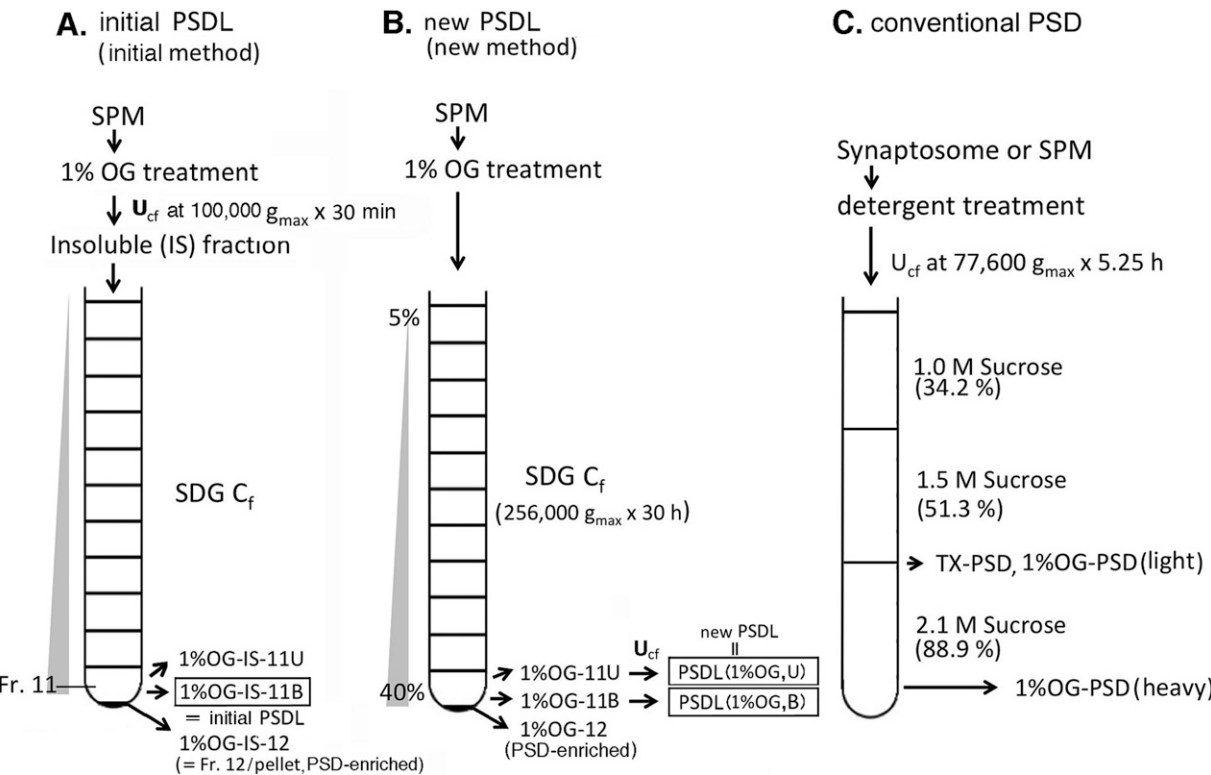

**Figure 1. Purification of PSDL preparations and PSDs.**
**(A, B)** Purification protocols for PSDL preparations and PSDs. We refer to 1% OG-IS-11B and PSDL (1% OG, U) as initial and representative new PSDL preparations, respectively. U and B refer to the upper and lower portions (see the Materials and Methods section). Fraction 12 (1% OG-IS-12 and 1% OG-12 in the initial and new methods, respectively, are pellet fractions and enriched in PSD [Suzuki et al, 2018]). **(A, B)** Ultracentrifugation ($U_{cf}$) was carried out before and after sucrose density gradient centrifugation ($C_f$) in (A) and (B), respectively. **(C)** Purification method for conventional PSDs (Suzuki et al, 2019). The conventional procedure used only TX-100. Two types of PSDs, 1% OG-PSD (light) and 1% OG-PSD (heavy), were obtained when OG was used in the conventional procedure. For TX-PSD, but not OG-PSDs, materials recovered at the interface between 1.5 and 2.1 M sucrose were treated with TX-100/KCl and spun down to obtain final TX-PSD, as in the original procedure (Cohen et al, 1977; Suzuki et al, 2019).

section). Thus, an additional ultracentrifugation step before the SDG ultracentrifugation step appeared to make some proteins insoluble in SDS, most likely because of the fact that highly concentrated conditions of a large variety of proteins around the PSDL caused extensive protein–protein interactions, resulting in protein aggregation. The amount of protein recovered in PSDL (1% OG, U) and PSDL (1% OG, B), PSDL preparations purified by the new method, were 12.9 ± 0.84 and 4.9 ± 1.95 µg (average ± SE, n = 3), respectively, from 3 mg of synaptic plasma membrane (SPM) protein. (See Fig 1B for PSDL preparations. To discriminate the PSDL preparations, codes showing OG concentration, location in the fraction 11 [upper, U, or bottom, B], and age of rats [7 d or 6 w] are supplemented. For simplicity, the supplementary codes 1% OG, U, and 6 w, either partly or all, are sometimes unwritten in the text; OG, n-octyl-$\beta$-D-glucoside.)

The presence of PSDL-like structures was checked by negative-staining EM in the 1% OG-11U and 1% OG-11U fractions (see Fig 1B for fraction name) without laborious ultracentrifugation before starting the expansion of this project. The structures contained in these fractions are not compressed by ultracentrifugation force, whereas these fractions contain soluble proteins and those solubilized with 1% OG. PSDL-like structures were confirmed,

together with a number of small structures (arrows) scattered widely (Fig 2A). Lattice-like structures in the 1% OG-11U were less dense than 1% OG-11B. The scattered small structures were greatly decreased after ultracentrifugation. We also confirmed the abundance of the PSDL-like structures similar to those prepared using the initial method (Suzuki et al, 2018) in other preparations (Fig 2B-a and B-b). Based on the results and the substantial similarities between the purification protocols, we refer to these preparations as new PSDL preparations. The structures in PSDL (1% OG, B) had a slightly higher protein density than those in PSDL (1% OG, U). We used PSDL(U) for the initial step experiments in this study based on the estimated substantial similarity and to avoid interference due to PSD/pellet contamination that tends to occur with the PSDL(B) samples.

Similar structures (possibly immature PSDL structures) were observed in the PSDL (1% OG, U, 7 d) and PSDL (1% OG, B, 7 d), which were purified from SPM prepared from 7-d-old rats (Fig 2B-c and B-d). Most of the structures in the PSDL (1% OG, U, 7 d) were not near-round in shape, different from typical mature PSDs (Fig 2C). Small globular structures of ~15 nm in diameter were abundantly associated with the PSDL (1% OG, B, 7 d) compared with PSDL (1% OG, U, 7 d).

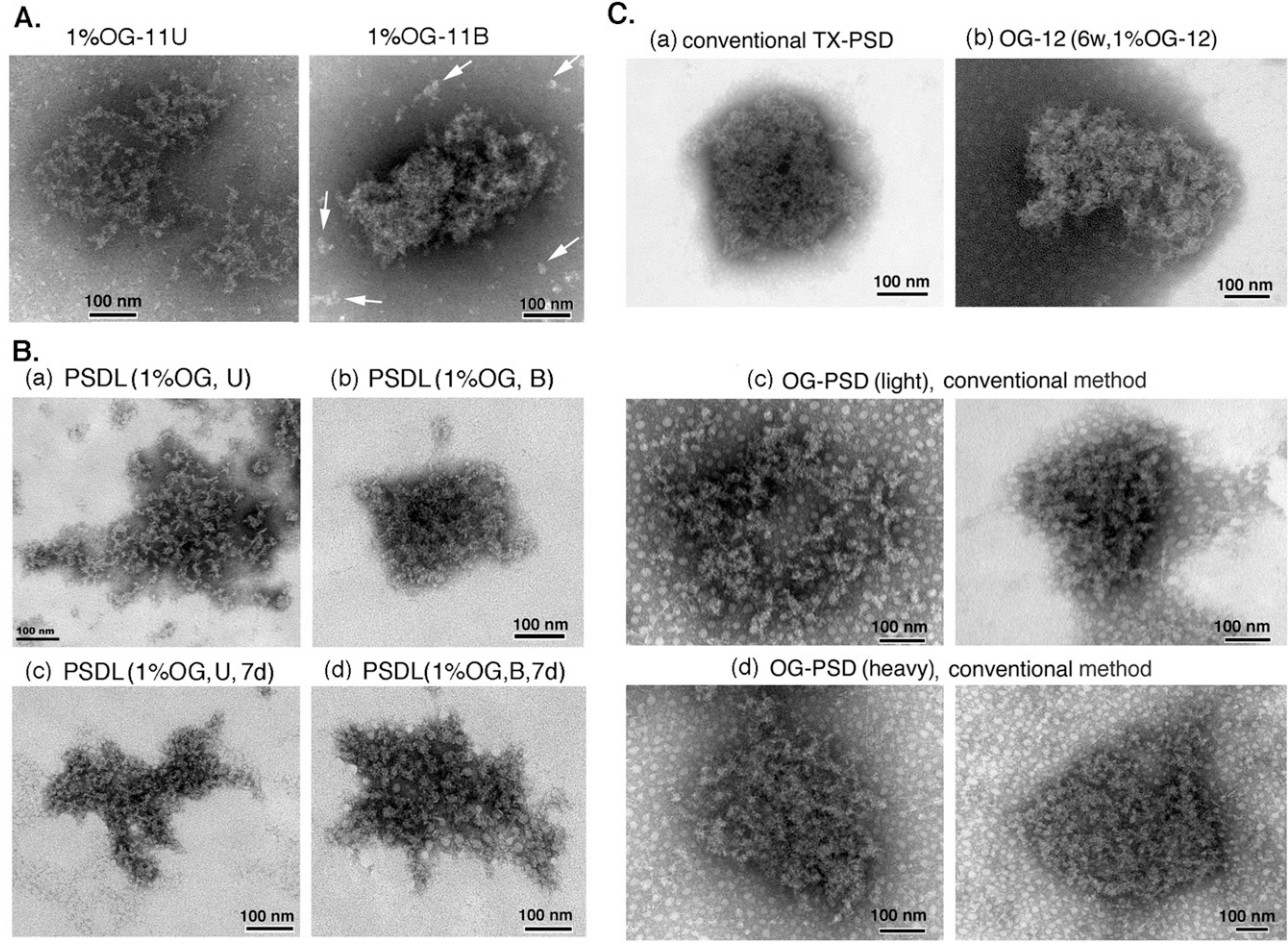

**Figure 2. Structures in the PSDL and PSD preparations examined by negative-staining EM.**
**(A)** Structures in the 1% OG-11U and 1% OG-11B fractions (see Fig 1B for fraction names). **(B)** Structures in the PSDL fractions examined. PSDL (1% OG, U) (B-a) and PSDL (1% OG, B) (B-b) were prepared from synaptic plasma membrane of 6-wk-old rats. PSDL (1% OG, U, 7 d) (B-c) and PSDL (1% OG, B, 7 d) (B-d) were prepared from 7-d-old rats. **(C)** Structures of various PSD preparations. Various PSD fractions were prepared by different protocols and examined by negative-staining EM to show the resemblance of "PSDL-like" structures to PSD. Representative examples are shown. **(C-a)** PSD prepared by the conventional method using TX-100 (Suzuki et al, 2019). **(C-b)** PSD contained in 1% OG-12. **(C-c, C-d)** PSD prepared by the conventional method, using OG instead of TX-100. Both OG-PSD (light) and OG-PSD (heavy) are shown. All PSD preparations were prepared from the forebrain of 6-wk-old rats. Scale bar, 100 nm.

The similarity of the meshwork structure to PSD was verified by comparing the purified PSDL with various types of PSD preparations (Fig 2C). The negative staining of various PSD preparations showed structures highly packed with molecules compared to the PSDL structures. Clear lattice-like structures were hardly observed in the PSDs, in particular, conventional PSD prepared by Triton X-100 (TX-100) treatment (TX-PSD) (Fig 2C-a). OG-insoluble PSDs were purified by the method shown in Figs 1B and 2C-b or following the conventional PSD purification method (Cohen et al, 1977; Suzuki et al, 2019) (Fig 2C-c and C-d). After the step-wise SDG ultracentrifugation of 1% OG-treated synaptosomes, opaque bands and aggregates appeared at the 1.5–2.1 M sucrose interface and in the 2.1 M sucrose layer. Both of these OG-insoluble materials consisted of PSD-like structures based on negative-staining EM observations (Fig 2C-c and C-d). Thus, two types of OG-PSDs (PSD prepared by OG treatment) were recovered: 1% OG-PSD (light) and 1% OG-PSD (heavy) (Fig 1C). The two structures were difficult to discriminate in negative-staining EM. However, 1% OG-PSD (heavy) appeared to

be slightly denser packed. Thus, in the case of 1% OG, the conventional PSD purification protocol produced two PSD preparations with different densities.

## Protein composition of PSDL preparations revealed by electrophoresis and Western blotting

Next, we examined the protein components of the new PSDL preparations (Fig 3A); hereafter, we will omit "new," except for some cases. The SDS–PAGE profiles of the PSDL (1% OG, U) and PSDL (1% OG, B) were different from the initial PSDL/1% OG-IS-11B, in which actin was almost the sole component (Suzuki et al, 2018). This may be due to improvements in the solubility to SDS of most proteins in the PSDL preparation. The protein profiles in SDS–PAGE were markedly similar between PSDL (1% OG, U) and PSDL (1% OG, B) (Fig 3A-a). The protein profiles of the PSDL were also similar between 6 w and 7 d (Fig 3A-c). In contrast, the SDS–PAGE profiles of the PSDL preparation were different from those of PSD (Fig 3A-a and A-b). The

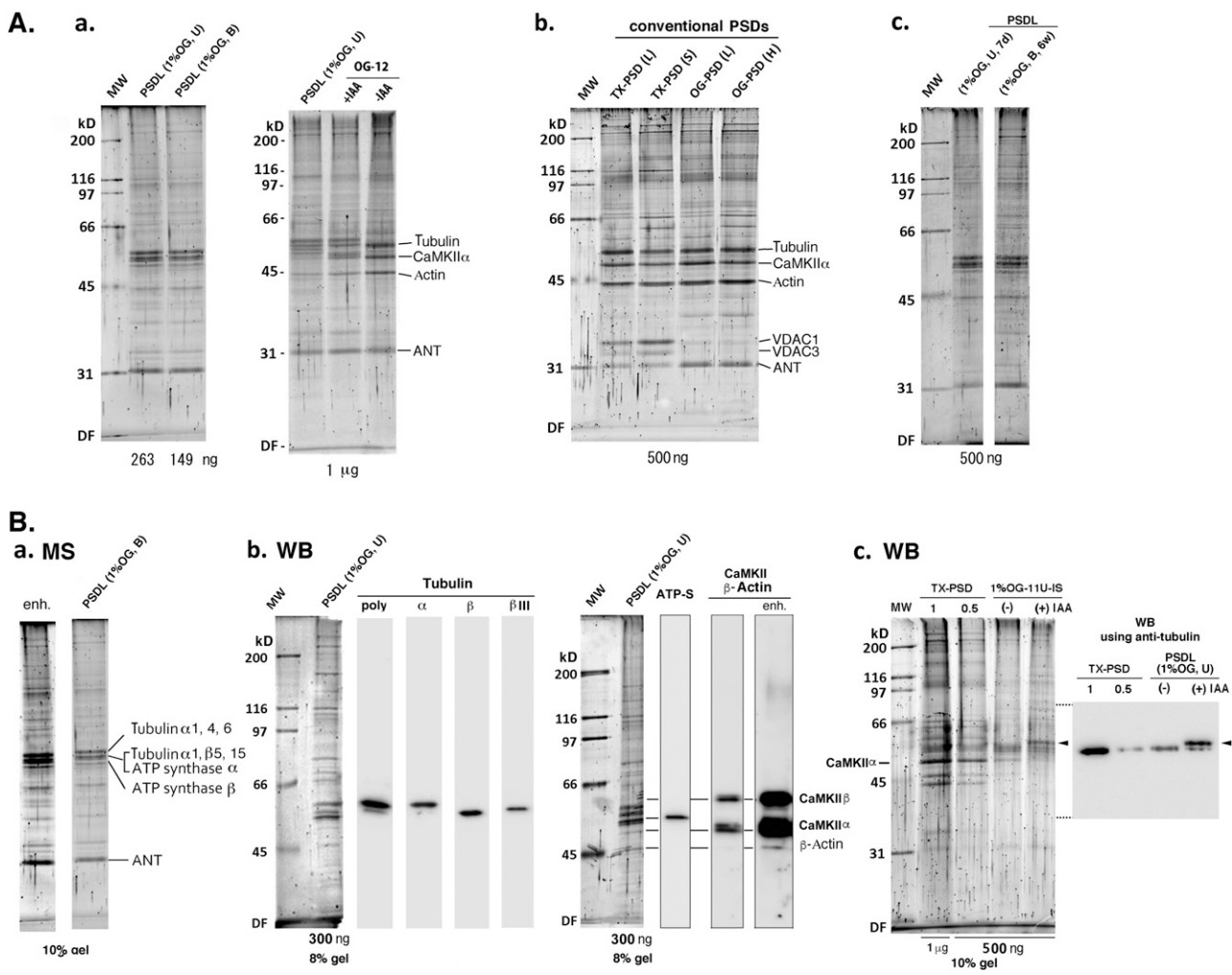

**Figure 3.  Analysis of protein components in the PSDL preparations by electrophoresis and Western blotting.**
**(A)** SDS–PAGE profiles of PSDL preparations and PSD preparations (OG-12, TX-PSDs, and OG-PSDs). Proteins were separated on a polyacrylamide gel (10%) and stained with SYPRO Ruby or Oriole. **(A-a)** Comparison of proteins in PSDL preparations and PSD. OG-12 was purified in the presence or absence of IAA. **(A-b)** Comparison of proteins in various PSDs. TX-PSD (L) and TX-PSD (S) are TX-PSDs prepared by long and short protocols via TX-100 treatment of synaptic plasma membrane and synaptosome, respectively (Suzuki et al, 2019). **(A-c)** Comparison of proteins in the PSDL preparation purified from immature (7 d) and adult (6 w) rat forebrain. Protein identification indicated in (A-a) and (A-b) is based on the previous identification by mass spectrometry (MS) (Liu et al, 2013; Zhao et al, 2014). PSDL (1% OG, U) and PSDL (1% OG, B) were prepared at least in triplicate, with essentially the same SDS–PAGE profiles. TX-PSD (S) and TX-PSD (L) were prepared from adult brains more than 120-times and 10-times, respectively, with essentially the same protein SDS–PAGE profiles. The obvious discrepancy between the protein amounts applied and the visualized band densities is due to differences in the conditions during capturing fluorescence. **(B)** Identification of major proteins in the PSDL preparations by Western blotting. Proteins were separated on polyacrylamide gel (10% or 8%) and stained with SYPRO Ruby or Oriole. **(B-a)** Protein identification by MS. Sypro Ruby-stained proteins were destained with $H_2O$, stained with silver, excised from electrophoretic gel, and identified by MS. The left lane shows the enhanced signal of the right lane, which is the same as the lane shown in (A-a). Protein IDs for tubulin *α*, P68366 and P68365; for *tubulin β*, P6987;for *ATP synthase α*, P19483 and Q9TM26; for ATP synthase *β*, Q05825; for ANT, P02722, Q05962, and P51881. **(B-b)** Identification of tubulin and other proteins by Western blotting. Tubulin subunits were detected with various anti-tubulin antibodies (polyclonal [poly], *α*, *β*, and *β*III). CaMKII and actin were detected with mixed antibodies containing anti-CaMKII*α*, anti-CaMKII *β*, and anti-actin antibodies. Western blotting was carried out two to three times with substantially the same results. A lane marked with enh. is an enhancement of the signals in the neighboring left lane to visualize the actin band. **(B-c)** Comparison of proteins in the preparations purified in the presence or absence of IAA. Protein staining (Oriole) and Western blotting using anti-tubulin polyclonal antibody are shown in the left and right panels, respectively. Scales of gel and blot were the same. Arrowheads indicate the same position. Protein amounts applied in each lane are shown below gels. MW, molecular weight standards. WB, Western blotting. ATP-S, ATP synthase. VDAC, voltage-dependent anion-selective channel protein.
Source data are available for this figure.

protein profile of OG-12, a type of PSD preparation (fraction 12 prepared from the OG treatment of SPM, Fig 1B), resembled that of conventional TX-PSD when prepared without iodoacetamide (IAA), whereas it appeared different when prepared with IAA (Fig 3A-a, right panel). Thus, the difference between PSD and PSDLs is likely

owing to protein modification by IAA, which changed the electrophoretic mobility. Considering the effects of IAA, the protein profile of PSDL is similar to that of PSD, except for a low content of *α* subunit of $Ca^{2+}$/calmodulin-dependent protein kinase II (CaMKII*α*) and actin. This result suggests that the subcellular structure, which was purified

by a newly developed method and termed the new PSDL, is closely related to the PSD.

The major proteins were identified by protein sequence analyses of the bands excised from the electrophoretic gel, as the $\alpha$ and $\beta$ isoforms of tubulin, ATP synthase $\alpha$, $\beta$, and ATP/ADP translocase (ANT) (Fig 3B-a). ATP synthase and ANT are typical mitochondrial proteins (Ko et al, 2003). The presence of tubulin $\alpha$, $\beta$, $\beta$III, CaMKII $\alpha$, $\beta$, and ATP synthase in the PSDL preparation was confirmed by Western blotting (Fig 3B-b). Tubulin was highly concentrated in the PSDL preparation compared with TX-PSD, which was confirmed by Western blotting (Fig 3B-c). The upward shift of tubulin band in the PSDL fractions prepared in the presence of IAA was also confirmed (Fig 3B-c). Actin was not a major component of the PSDL preparation (Fig 3B-b).

Next, the PSDL preparations were purified using 5% OG and 0.75% OG, in addition to 1% OG, to search for proteins that play key roles in the formation of the backbone structure of the PSDL and those associated with the backbone structure, respectively. The concentrations of detergent used were above the critical micelle concentration of OG (20–25 mM [0.585–0.7%]). We selected 0.75% OG, which was slightly above the critical micelle concentration of OG, to isolate the structures in which PSDL-associating proteins were expected to be abundantly associated. Five percent OG is practically the maximum concentration that can be used. We examined the morphological differences between these three PSDL preparations: all three preparations preserved PSDL-like meshwork structures (Fig 4A). However, the structures appeared sparser, and the meshwork structure was more clearly visible in the sample prepared after treatment with a higher detergent concentration (Fig 4A). In contrast, structures that were not observed in the 1% OG and 5% OG samples were associated with the lattice structures in the 0.75% OG sample (arrows in Fig 4A). Based on these morphological differences, we tentatively grouped the PSDL preparations into "lean" (1% OG and 5% OG) and "enriched" types (0.75% OG).

Protein recovery of the PSDL preparations increased with detergent concentration (Fig 4B). The protein recovery of the PSDL preparations were calculated to be 0.3–0.9% of SPM, whereas that of PSD is 4.2% (Suzuki et al, 2019). Differences in the protein composition among these three preparations were not significant, except for small differences in some bands including those corresponding to CaMKII$\alpha$ (Fig 4C). In contrast, changes in content were visible in Western blotting (Fig 4D). The concentration of tubulin was relatively unchanged, and was maintained at a high level in the three PSDL preparations. Tubulin was found to be more concentrated in the PSDL preparations than in TX-PSD (Fig 4D and E), whereas the reduction was unclear in the OG-12 (Fig 4F), another type PSD preparation. The relatively low content of tubulin in the TX-PSD, but not in OG-12, compared with the PSDL preparations may depend on the detergent used because tubulin in the PSD is more resistant to extraction by OG than by TX-100 (Somerville et al, 1984). The ATP synthase content was also maintained in the three PSDL preparations. However, the ATP synthase content in the PSD, both TX-PSD and OG-12, was low (Fig 4D–F).

Other proteins examined decreased as OG concentration increased, reaching minimum levels or falling to levels below the detection level in the PSDL (5% OG). $\alpha$-internexin ($\alpha$-IN), Homer1, and actin were extremely reduced or below the detection level in

the 500 ng proteins of the PSDL preparations, despite their presence in the PSD preparation. Shank1 was found to be below the detection level in the Western blotting using 500 ng proteins of these three types of PSDL preparations.

## Analysis of protein components in three different PSDL preparations by proteomics method

The proteins in the lean-type PSDL/PSDL (1% OG), enriched-type PSDL/PSDL (0.75% OG), and initial PSDLs/1% OG-IS-11B (Suzuki et al, 2018) were comprehensively identified by MS using the shotgun method. Both MPEX PTS-soluble and insoluble fractions were investigated (see the Materials and Methods section and Fig S1A). The protein components in the enriched-type PSDL (Table S1), the lean-type PSDL (Table S2), and the initial PSDL were compared. The shotgun analysis of the initial PSDL was used to efficiently narrow down the number. The results are shown in a Venn diagram (Fig 5).

The enriched-type PSDL preparation mostly contained the components of the lean-type PSDL, whereas relatively few proteins of the initial PSDL were components of the enriched PSDL (Fig 5A and B). In other words, the initial PSDL preparation contained a large number of proteins that were absent in the new PSDL preparation. These proteins may be artificially associated with the PSDL structure under forced concentration by ultracentrifugation in the initial purification protocol. Thus, 102 components of the initial PSDL (Fig 5A) may originate from subcellular components unrelated to the PSDL structure in vivo. Proteins common to these three preparations (130 proteins) may contain key proteins for the PSDL structure because all of these preparations have PSDL structures. Furthermore, candidate essential structural proteins for the PSDL structure could be confined to 58 common proteins in the MPEX PTS-insoluble fractions of these three preparations supposing that PSDL structures are not completely solubilized with MPEX PTS under the conditions used (Fig S10 [Suzuki et al, 2018]). Thus, we tentatively termed this group of proteins as the minimum essential cytoskeleton (MEC) proteins (Fig 5C). We also termed the other components of the enriched-type PSDL non-MEC components (Fig 5C). Non-MEC proteins are those that are relatively weakly associated with the PSDL structure and can be easily dissociated from the PSDL structure by increasing the detergent concentration. Non-MEC proteins may be subordinate associated proteins to the PSDL structure.

The protein categories contained in the MEC and non-MEC groups are summarized in Table 1, with the numbers of protein species in each protein category. The representative structural or structure-related proteins in the MEC are PSD-95, tubulin, spectrin, actin, CaMKII, and Hsc71 (Tables 1 and S3). These proteins may play key roles in constructing the basic structure of the PSDL. Channels/transporters, presynaptic proteins, glial proteins, primarily mitochondrial proteins, and proteins involved in membrane trafficking/secretion/cell fusion, ubiquitination, oxidation/reduction, metabolic pathways, and protein synthesis are also listed as MECs. However, these are not structural proteins; therefore, they may not play key roles in the backbone of the PSDL structure, and are prevalent in the non-MEC group (Table 1).

Proteins in the non-MEC group are listed in Table S4. They include PSD scaffold/adaptor proteins other than PSD-95, neurotransmitter

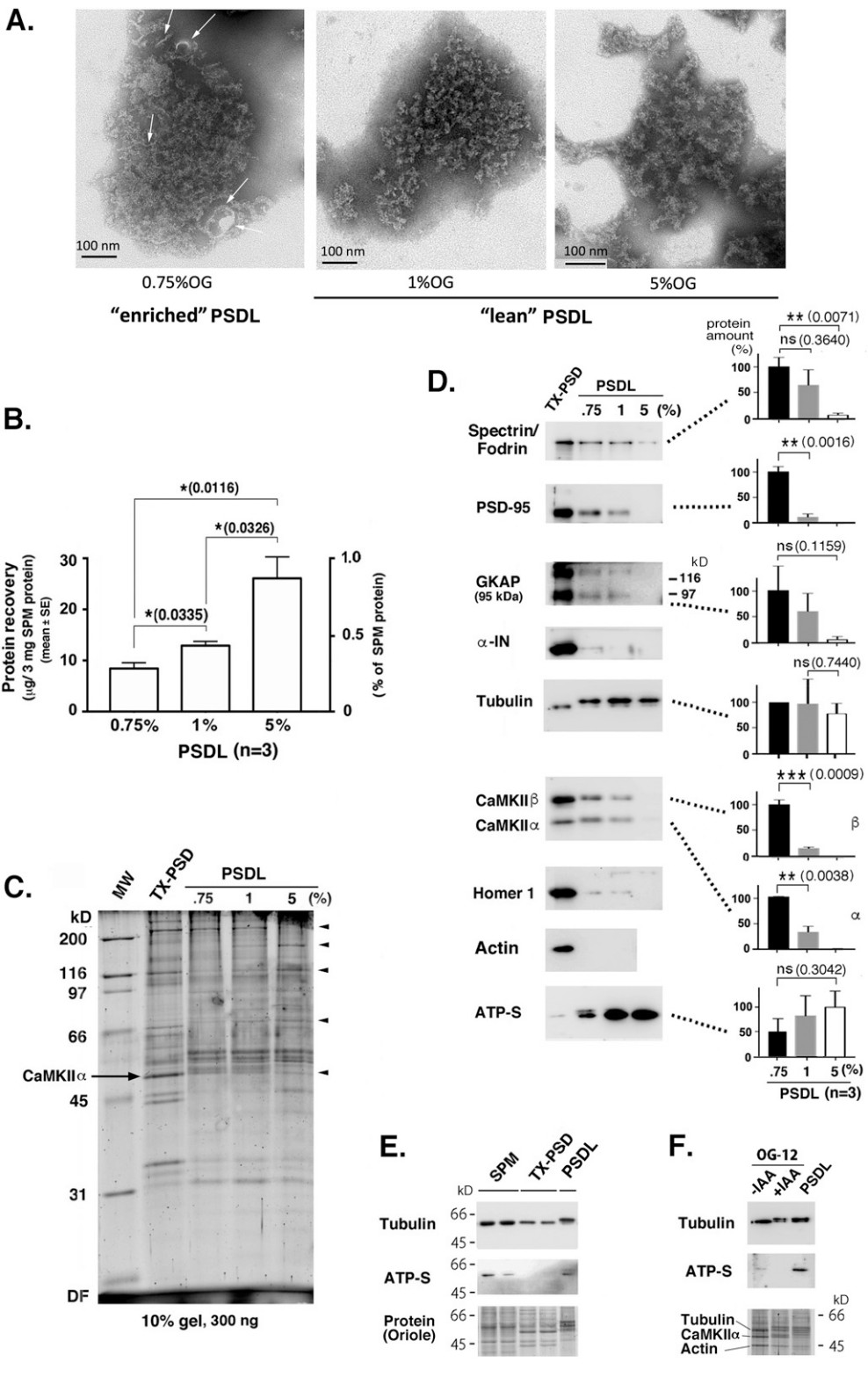

**Figure 4. Analysis of the enriched- and lean-type of PSDL preparations.**
**(A)** Electron microscopic observation of the enriched- and lean-type of PSDL structures. PSDL preparations were prepared using 0.75%, 1%, or 5% OG for synaptic plasma membrane (SPM) prepared from the forebrain of 6-wk-old rats. The structures contained in these preparations were examined by negative-staining EM. Representative images are shown. These PSDL structures are tentatively classified into "enriched" and "lean" types based on morphology by placing a border between the 0.75% OG and 1% OG preparations. This was supported by further experiments using Western blotting and mass spectrometry. Arrows on the PSDL (0.75% OG) structure indicate membrane-like and other structures, which were not present in the lean-type PSDL preparations. Scale bar, 100 nm. For more views of the enriched PSDL structure, see Fig 6C. **(B)** Protein recoveries (mean ± SE) of the three PSDL preparations obtained after treatment of SPM with 0.75%, 1%, or 5% OG. **(C)** Protein profiles of the three preparations stained with Oriole. Arrowheads indicate the bands of which densities were changed depending on the concentration of OG. These changes were observed in repeated electrophoresis. The CaMKIIα band in the PSD is indicated. **(D)** Contents of typical PSD proteins in the "lean" and "enriched" PSDL preparations. Western blotting was carried out to measure the protein amounts. For quantification, three different preparations for 0.75% OG, 1% OG, and 5% OG (total nine preparations) were used. Results (mean ± SE) are shown on the right. The protein amount of each protein was normalized to tubulin in each preparation, and plotted by setting the amount in the PSDL (0.75% OG) as 100%, except for ATP-S. For tubulin, percentages to the PSDL (0.75% OG) tubulin are plotted. **(E)** Western blotting of tubulin and ATP synthase (ATP-S) in the SPM, PSD (TX-PSD), and PSDL (U). **(F)** Western blotting of tubulin and ATP-S in the PSD (OG-12), and PSDL(B). OG-12 fractions were prepared in the presence or absence of IAA. Entire protein profile of (F), but left-right reversed, is shown in Fig 3A-a. P-values (t test, n = 3) are indicated in parentheses. For significance code, see Fig 6B. ns, nonsignificant. **(C, D, E, F)** Protein amounts applied for each lane: 300 ng (C), 500 ng (D), 5 μg (E), and 1 μg (F). Source data are available for this figure.

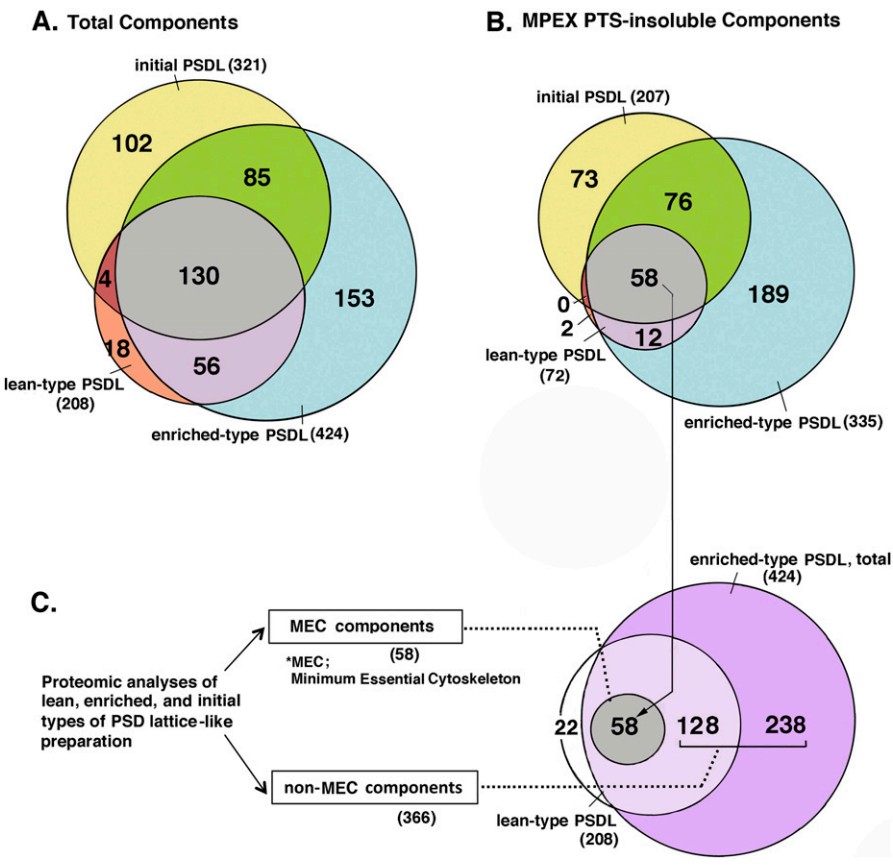

**Figure 5. Analysis of protein components identified by comprehensive mass spectrometry.**
Proteins in the three PSDL preparations: initial PSDL/1% OG-IS-11B (Suzuki et al, 2018), lean-type PSDL (1% OG, U), and enriched-type PSDL (0.75% OG, U), respectively, were recovered by the chloroform/methanol protocol, and solubilized with MPEX PTS reagent. Proteins in both MPEX PTS-solubilized and insoluble fractions were comprehensively identified by mass spectrometry using the shotgun method (Fig S1A). Protein distribution in these samples was compared with Thermo Proteome Discoverer. **(A, B)** Comparison of total proteins (A) and MPEX PTS-insoluble components (B) among the three preparations are shown in the Venn diagram. **(C)** Grouping of the PSDL components into the minimum essential cytoskeleton (MEC) and non-MEC components. **(B)** MEC is the same as the gray area that contains 58 protein species in (B). The protein numbers in each area are indicated in parentheses. For protein components of MEC and non-MEC, see Tables S3 and S4.

receptors, cytoskeletal components other than tubulin, and proteins related to cellular signaling. Scaffold and adapter proteins other than PSD-95, such as chapsyn110, SAP102, Dlgap1/GKAP/SAPAP1, Dlgap2, Dlgap3/SAPAP3, Dlgap4, ArgBP2, Begain, Caskin1, Densin-180, Disable homolog-2-interacting protein, Homer1, Homer2, leu-rich proteins, Lin-7 homolog C, Shank1, Shank2, and Shank3, are grouped into non-MEC.

Other cytoskeletal and cellular structure-related proteins, such as cell adhesion molecules, junctional proteins, microtubule (MT)-related proteins, $\alpha$-IN, neurofilament proteins, spectrin-related proteins, actin regulatory proteins, myosin, and septins are also excluded from the MEC (Table 1) and may not be involved in the construction of the PSDL backbone structure. PSD proteins, such as neurotransmitter receptors, those related to signaling processes, and most membrane trafficking-related proteins are also listed as non-MEC (Table 1). They may function as proteins necessary for PSD functioning, and thus have been associated with the PSDL structure.

The amount of proteins of structural interest, such as tubulin, other cytoskeletal proteins, and typical scaffold/adaptor proteins in the PSDL (1% OG) preparation were estimated by exponentially modified protein abundance index (emPAI), which is an expedient method to estimate the absolute protein amount based on proteomics data (Ishihama et al, 2005). All cytoskeletal and scaffold/adaptor proteins in PSDL (1% OG) were selected. The results suggest an abundance of tubulin, followed by CaMKII, actin, and PSD-95 (Fig S2). The amount of $\alpha$-IN and other cytoskeletal proteins, including

actin-related and scaffold/adaptor proteins (other than PSD-95), were estimated to be lower than those of tubulin and CaMKII. The amounts of ATP-synthase $\alpha$ and ANT1/2, typical mitochondria-residing proteins, in the PSDL (1% OG) preparation were estimated to be as large as the CaMKII$\alpha$ and tubulins, respectively, by emPAI.

## The distribution of protein components on purified PSDL structures observed by immunogold negative-staining EM

We then investigated the localization of tubulin molecules on the PSDL structure in PSDL (1% OG) using immunogold negative-staining EM. Tubulin was widely distributed throughout the PSDL structure (Fig 6A-a and A-b), suggesting that tubulin is a predominant constituent of the PSDL structure. We further examined the distribution of tubulin $\alpha$, $\beta$, and $\beta$III isoforms (Fig 6A-c–A-e) and confirmed the presence of these three tubulin isoforms on the PSDL structures. Tubulin $\beta$ tended to be distributed widely throughout the PSDL structure, whereas tubulin $\alpha$ and $\beta$III showed a more limited distribution. The random distribution of tubulin $\alpha$ and tubulin $\beta$/$\beta$III on the PSDL structure was revealed by double immunogold labeling (Fig 6A-f). The number of immunogold particles per square micrometer for tubulin (polyclonal) was close to the sum of those for $\alpha$, $\beta$, and $\beta$III (Fig 6B). Tubulin-immunoreactive gold particles were also detected on the PSDs prepared by various methods (Fig S3), although the signals were not abundant on the

**Table 1.  Comparison of protein categories identified in minimum essential cytoskeleton (MEC) and non-MEC.**

| Protein category | No. of proteins | |
|---|---|---|
| | MEC | Non-MEC |
| PSD-95 | 1 | |
| Other Dlg family members | | 6 |
| Other Scaffold/Adaptor proteins | | 14 |
| Cell adhesion proteins | | 9 |
| NT receptors, other receptors, their associated proteins | | 13 |
| Junctional proteins | | 3 |
| Gephyrin | | 1 |
| Tubulin | 7 | |
| Microtubule-related | | 11 |
| α-internexin | | 1 |
| Neurofilament | | 3 |
| Spectrin, spectrin complex | 1 | 7 |
| Dynein-related | | 1 |
| Kinesin, kinesin-like | | 2 |
| Actin | 1 | 1 |
| Actin regulation | | 20 |
| Myosin, myosin-related | | 5 |
| Septins | | 5 |
| CaMKII | 4 | |
| Channels, transporters | 5 | 23 |
| G-protein related | | 4 |
| Small G proteins, their regulators, downstream | 3 | 27 |
| Kinases, phosphatases, their regulators (except for CaMKII) | | 18 |
| Membrane raft | | 7 |
| Membrane trafficking, secretion, cell fusion | 2 | 18 |
| Heat shock proteins, chaperones | 2 | 11 |
| Transcription factors, nuclear proteins | | 8 |
| Ubiquitination | 1 | 1 |
| Extracellular proteins, secreted proteins | | 5 |
| Presynapitc proteins | 3 | 13 |
| Glia, myelin | 2 | 2 |
| Oxidation, reduction | 1 | 4 |
| Metabolic pathway | 2 | 9 |
| Primarily mitochondrial | 20 | 64 |
| Protein synthesis | 3 | 40 |
| Others | | 10 |
| | 58 | 366 |

Mass spectrometry analyses were carried out to identify protein components of the three types of PSDL preparations: enriched-type/PSDL (0.75% OG), lean-type/PSDL (1% OG), and initial one/1% OG-IS-11B. Proteins identified in the three preparations were grouped into minimum essential cytoskeleton (MEC) or non-MEC components (Fig 5C). Proteins are clustered into various categories of interest. The number of protein species in each category is listed. Keratins and trypsin are excluded from the list. NT, neurotransmitter. For the protein species in each category, see Tables S3 and S4.

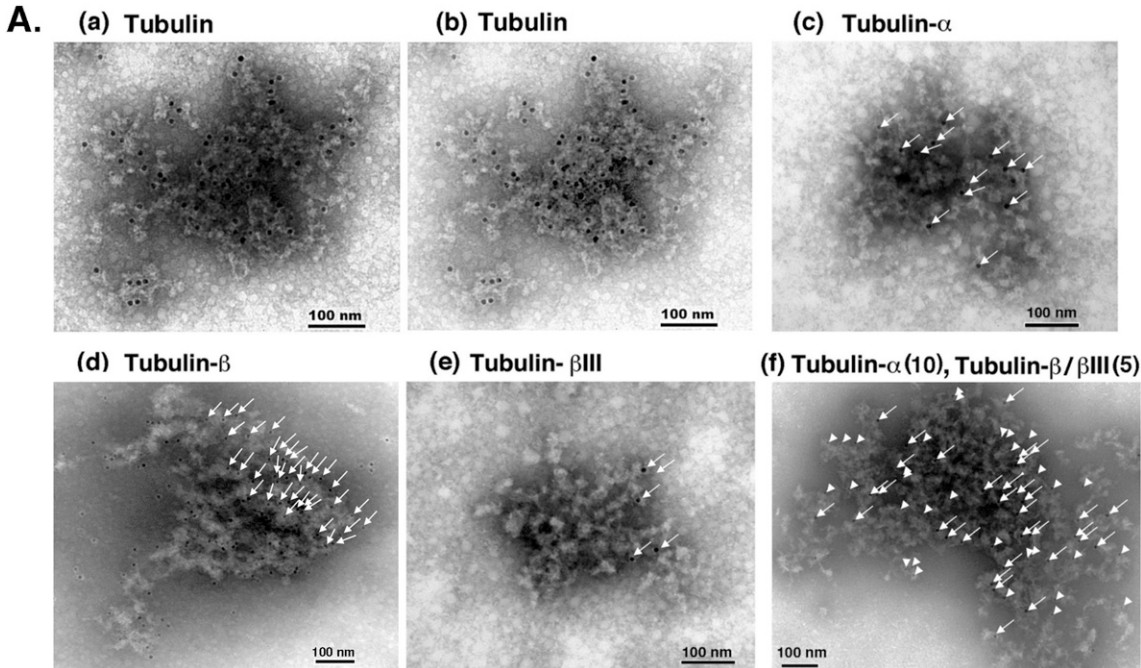

**Figure 6. The distribution of typical PSD proteins on the PSDL structure.**
**(A)** Distribution of tubulin-immunoreactive proteins on the PSDL structure. Immunogold negative staining was used to detect the distribution of tubulin in the PSDL (1% OG) structure. **(A-a, A-b, A-c, A-d, A-e)** The samples were spotted on carbon-coated formvar membrane on the EM grid, labeled with anti-tubulin polyclonal antibody (A-a, A-b) or monoclonal antibodies specific for tubulin α (A-c), tubulin β (A-d), and tubulin βIII (A-e), followed by gold (10 nm)-labeled secondary antibodies, and negatively stained. **(A-a)** and **(A-b)** are the same pictures with different whiteness to clearly show the mesh-like structure and gold particles, respectively. Gold particles are not indicated in (A-a) and (A-b) to avoid obstructing mesh or lattice-like structures, and those in (A-d) are indicated by arrows only in the upper right area. Samples were incubated with a mixture of anti-tubulin α, anti-tubulin β, and anti-tubulin βIII antibodies, and tubulin α and tubulins β/βIII were labeled with 10 nm (arrows) and 5 nm (arrowheads) gold particles, respectively, in (A-f). Scale bar, 100 nm. **(B)** Quantitative analyses of typical PSD proteins in the enriched-type and lean-type PSDL structures. Quantitative data (mean ± SE) of representative PSD scaffold/adaptor and cytoskeletal proteins. Immunogold labeling on PSDL (0.75% OG) and PSDL (1% OG) was counted and numbers of gold particles per $1\ \mu m^2$ are plotted. For tubulin, signals on PSDL (5% OG) were also examined. Significance was examined either by $t$ test (St) or Mann–Whitney's U test (U) depending on the normality of the sample distribution. $P$-values are indicated. Samples not significantly different from the negative controls are marked with ┼. Sample numbers are indicated in the parenthesis at the bottom. ns, nonsignificant. ATP-S, ATP synthase. NC, negative control; Rab, rabbit; Mo, mouse; Go, goat (animals in which primary antibodies were raised). **(C)** Distribution of structure-related PSD proteins on the PSDL (0.75% OG) and PSDL (1% OG) structures. Representative examples are shown. There are no tubulin-immunogold signals on the structure shown in (C-b) (tentatively named as "fine type"), and this type of structure was excluded from the quantitative analysis. Arrows indicate all gold particles (10 nm). Arrowheads indicate structures that were not found in the lean-type PSDL structures (not all are indicated, in particular in (C-g)). Homer1-immunostaining shown in (C-g) is excluded from the quantitative analysis because a different PSDL (0.75% OG) preparation was used for the quantitative analysis. Gold particles were not indicated in (C-k) because they are clearly identifiable. Scale bar, 100 nm. Source data are available for this figure.

PSDs compared with those on the PSDL, possibly because of the reduced accessibility of the antibody to tubulin in the PSDs due to the densely packed proteins.

Immunogold negative-staining EM was also used to examine the distribution of other cytoskeletal and scaffold/adaptor proteins identified as PSDL components (quantitative data are shown in Fig 6B and typical examples of the distribution of immunogold labels for each protein are shown in Fig 6C-a–6C-l). Tubulin was the most abundant among the proteins examined in both PSDL (0.75% OG) and PSDL (1% OG), and was abundant in the PSDL (5% OG) structures, followed by CaMKIIα and β. CaMKIIβ was significantly higher than the CaMKIIα signals in PSDL (0.75% OG). Two PSDL structures with extremely high content of CaMKIIβ were found (Fig 6B). PSD-95 appeared to be the most abundant among the scaffold/adaptor proteins examined, although there were no statistically significant differences between PSD-95 and Homer1. Homer1 was significantly higher than Shank1 and GKAP in PSDL (0.75% OG) but not PSDL (1% OG). Although both Shank1 and GKAP were very low, the GKAP levels

were significantly higher than the Shank1 levels in PSDL (0.75% OG), but not in PSDL (1% OG). The α-IN and actin levels were also very low in both PSDL (0.75% OG) and PSDL (1% OG). The representative mitochondrial protein ATP synthase 5A1 was at nearly the same level as PSD-95 and Homer1. ANT was nearly absent in the PSDL structure, although nonspecific clustered gold particles that appeared because of anti-goat IgG secondary antibody were found in some areas. A relatively deviated abundance of immunogold particles was observed for PSD-95, Homer1, CaMKIIα and β, in contrast to Shank1, α-IN, and actin. Overall, these proteins were not present in the sparse area on the PSDL structures. This phenomenon may be due to the disparity in the solubilizing process of individual PSDL structures, suggesting the subordinate association of these proteins in vivo with the backbone structure of the PSDL rather than the structural role.

Fibrous, membrane-like, or globular structures, which were not found in the lean-type PSDL structures, were frequently associated with PSDL structures in the enriched-type PSDL preparation (arrows

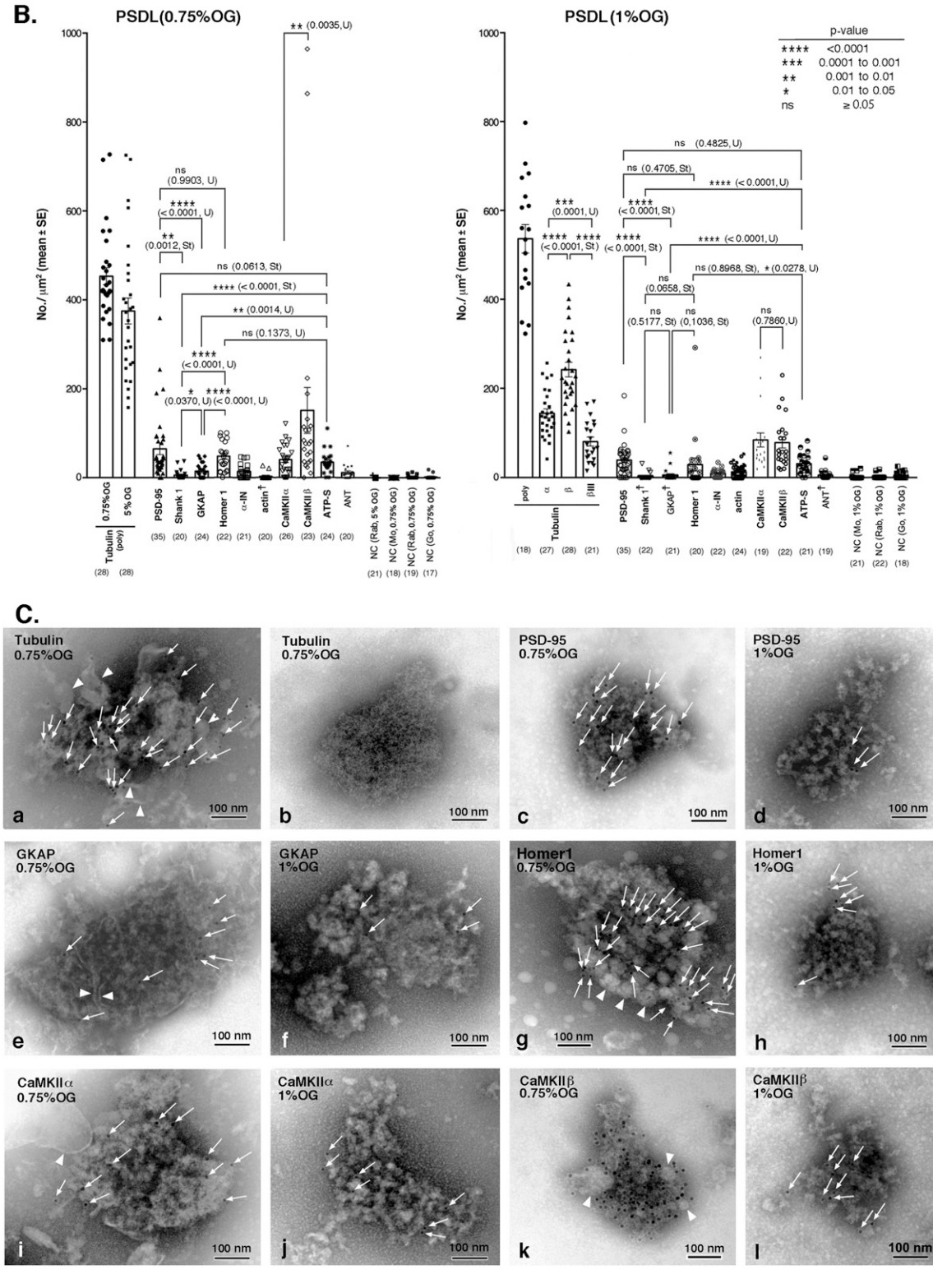

**Figure 6. Continued**

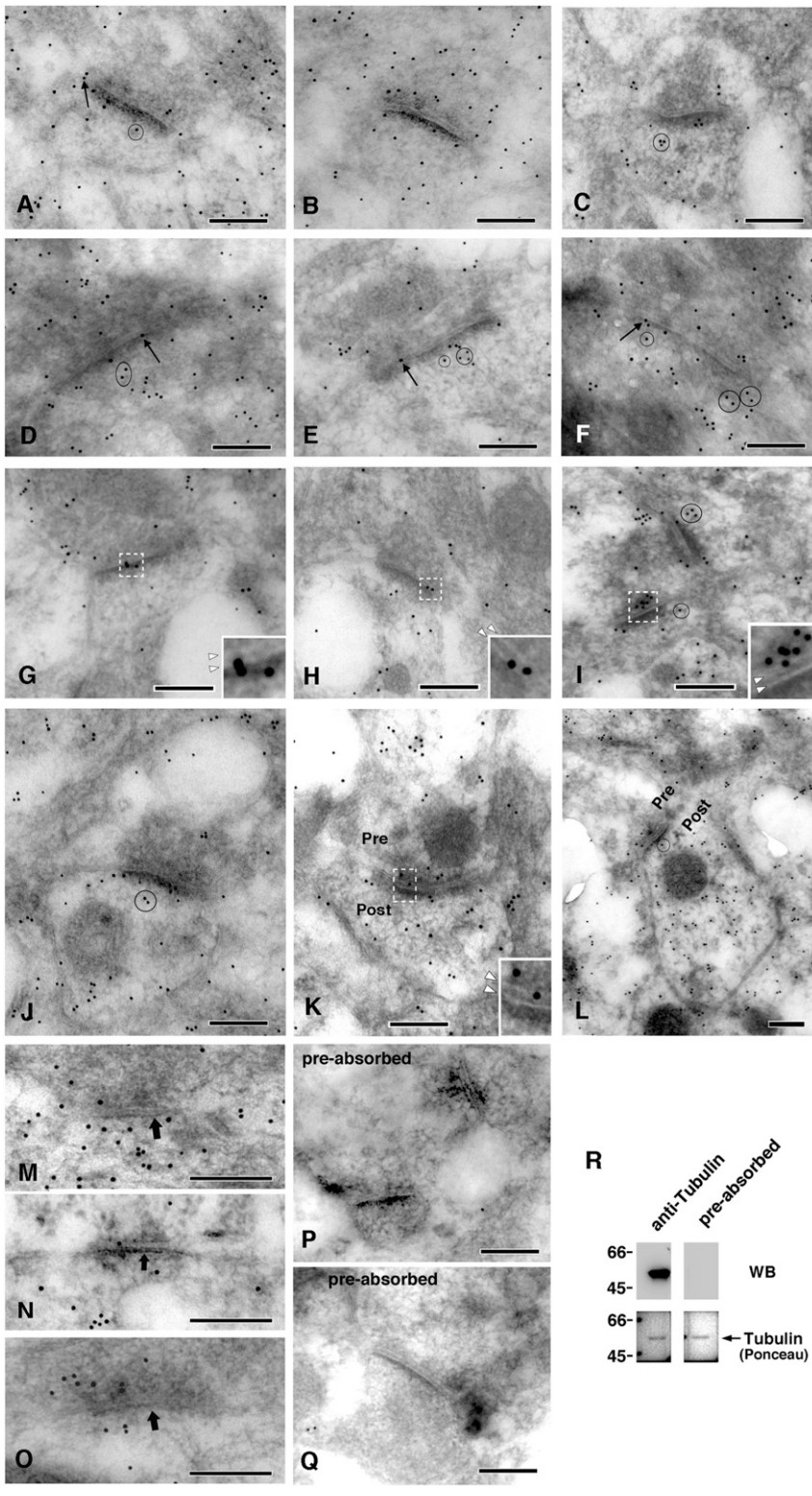

**Figure 7. Localization of tubulin immunoreactivities in the synaptic areas in the mouse brains.**
The presence of tubulin was investigated in the mouse cerebral cortex by post-embedding labeling using anti-tubulin antibody followed by immunogold (10 nm) labeling. **(A, B, C, D, E, F, G, H, I, J, K, L)** Typical examples of type I excitatory synapses with PSD cores labeled with multiple immunogold particles. Arrows indicate gold particles immediately below the plasma membrane in the synaptic zone. Typical gold particles localized in the PSD pallium regions are surrounded by circles. **(G, H, I, K)** Enlarged images of areas surrounded by white broken lines are inserted in (G, H, I, K), where white arrowheads indicate unstained pre- and post-synaptic plasma membranes (upper and lower arrowheads indicate pre- and post-synaptic membranes, respectively). **(K, L)** Presynaptic and postsynaptic sides (pre and post, respectively) are indicated in (K) and (L). **(M, N, O)** Probably type II inhibitory synapses judged by their thickness. These are indicated by the thick arrows. Note that these photos are enlarged more than the other photos. **(P, Q)** Negative controls stained with an anti-tubulin antibody preabsorbed with purified tubulin. No immunogold was detected in either of the PSDs. Scale bar, 200 nm. **(R)** Activity of anti-tubulin antibody (polyclonal, rabbit) preabsorbed with purified tubulin. Tubulin purified from porcine brains was applied to each lane (0.5 g), and only the tubulin-containing region was used for Western blotting. Western blotting using the anti-tubulin antibody and the preabsorbed antibody was carried out simultaneously under the same conditions. Western blots (WB) and corresponding sheets stained with Ponceau are shown. The contrast was adjusted under the same conditions between the experimental and negative control groups.

Source data are available for this figure.

in Fig 4A and arrowheads in Fig 6C). This morphological difference is in agreement with the finding that enriched-type preparations contained many non-MEC proteins in addition to the MEC proteins. Enriched-type PSDL structures with abundant PSD-95, Homer1, CaMKIIβ were found (Fig 6C-c, 6C-g, and C-k, respectively). The highly deviated concentration of these proteins suggests their weak binding with the backbone structure of the PSDL as mentioned above.

### The distribution of tubulin in the synaptic areas in the brain

Post-embedding immuno-labeling EM using immunogold was carried out to examine the presence and distribution of tubulin in the PSD in situ. Perfusion and block fixation were performed at room temperature to minimize tubulin exodus from synapses (Cheng et al, 2009a). Fixation with osmium was not performed to minimize the loss of antigenicity and morphological damage (Valtschanoff & Weinberg, 2001) and, owing to this omission, membranes, such as plasma membranes and those of synaptic vesicles, were not stained. We used LR-gold resin, which is superior for the preservation of morphology and antigenicity (Migheli et al, 1992). Samples were counterstained with uranium to avoid heavy staining of the PSD, which would prevent the easy identification of immunogold particles. Negative controls (NCs) without primary antibody showed no immunogold particles (not shown). Another NC, using preabsorbed anti-tubulin antibody (Fig 7R), showed only trace amounts of immunogold (Figs 7P and Q and 8). Therefore, immunogold staining was determined to be specific for tubulin. We focused on the synapses and their neighbors, although tubulin-immunoreactive signals were distributed widely in various parts of neuronal cells, including MTs in dendrites and axons. Synapses with multiple-labeled PSD cores are shown in Fig 7. In addition to the labels inside and closely attached to the PSD core, immunogold was localized on the plasma membrane in the synaptic active zone, synaptic cleft (Fig 7G–I and K), regions immediately underneath the postsynaptic plasma membrane (arrows in Fig 7), and assumed PSD pallium regions (Fig 7A, C–F, and J–L). Postsynaptic area at a distance from the PSDs was also positive for the label. In some synapses, immunogold was distributed widely in presynaptic terminals (Fig 7B), spine heads (Fig 7C, J, and L), and both (Fig 7F and K). Immunogolds were also localized in the synaptic areas of type II inhibitory synapses (PSDs indicated by thick arrows, Fig 7M–O).

The distribution of tubulin-immunogold in the synaptic areas was quantified (Fig 8). 55 and 29 excitatory synapses were examined in the experimental and NC groups using the anti-tubulin antibody and the preabsorbed antibody, respectively. Type II synapses were not counted. Immunogold was significantly present in the above-mentioned subregions surrounding type I synapses on both presynaptic and postsynaptic sides (Fig 8A). Fifty-one percent of PSD cores were labeled with 1–8 immunogold particles per single PSD core (Fig 8A). Eighty-seven percent of the assumed PSD pallium regions were labeled (circles in Figs 7 and 8A). The density of immuno-labeling was significantly higher in the PSD core than in other synaptic subregions, except for the assumed PSD pallium (Fig 8B). Only a few labels were observed in the SPM and synaptic cleft; thus, statistical significance could not be determined; however, this is a good contrast that there were no labels in these sites of the NC tissue.

## Discussion

The PSDL was first identified in the 1970s as a deoxycholate (DOC)-insoluble PSD. However, the key molecules involved in the construction of the PSDL, and the way of its construction have not yet been specified, because the old day's PSDL (DOC-insoluble PSD) contained many proteins like conventional TX-PSD (Matus & Walters, 1975; Blomberg et al, 1977; Matus & Taff-Jones, 1978; Matus, 1981). Although we previously reported on the purification of the PSDL structure (Suzuki et al, 2018), the PSDL purified by the initial method was unsatisfactory for the analysis of the protein components. The robust concentration of the PSDL structure after ultracentrifugation in the presence of a large number of non-neighboring proteins immediately after detergent treatment of the SPM appears to induce their association with the PSDL structure. This results in the production of SDS-/mercaptoethanol-insoluble aggregates. In this study, we changed the order of ultracentrifugation and SDG ultracentrifugation steps (Fig 1A and B) to establish a new purification protocol that would avoid this undesirable association.

The electrophoretic protein profile of the PSDL preparation using the new method was different from the profile of other group's PSDL/DOC-insoluble PSD (Matus & Taff-Jones, 1978). This may be because the purified new PSDL structures were slightly lighter than PSD, and that some undetermined properties of OG were advantageous for the purification of the PSDL. We purified the SPM and PSDLs in a solution containing 2 mM IAA to prevent artificial protein oxidation (ex. S–S cross-bridge), which affects the resistance to extraction by detergent (Sui et al, 2000; Suzuki et al, 2019). It is also possible that the PSDL structure purified in this study may be a nascent structure that grows into PSD rather than a structure derived from a completed PSD.

After purifying a new PSDL preparation, we acquired information on its components and established a new PSDL model consisting of a non-MT tubulin-based backbone and its associated proteins. The PSDL has the following notable properties: (1) tubulin is by far the most common component; (2) it contains a minimum amount of other cytoskeletal proteins, such as actin, spectrin, and $\alpha$-IN; and (3) CaMKII and the major scaffold/adaptor proteins examined can easily be dissociated from the PSDL by relatively harsh detergent treatment (Fig 9). These findings suggest that the PSDL backbone is constructed and maintained without the typical PSD scaffold/adaptor proteins, actin, spectrin, $\alpha$-IN, and CaMKII.

### Tubulin is a major MEC protein

Proteomic analyses identified 58 proteins as MEC proteins (Table S3 and Fig 5), which may be key in constructing the backbone of the PSDL. Among these, tubulin is the most promising candidate molecule. Both $\alpha$ and $\beta$ tubulin isoforms were present in the PSDL structure. The abundance of tubulin in the PSDL preparation was outstanding compared with the other MEC components (Figs 3B and S2). Tubulin-immunogold particles were distributed widely in the purified PSDL structure (Fig 6A). Tubulin content was consistently high in the PSDL (0.75% OG), PSDL (1% OG), and PSDL (5% OG) preparations (Fig 4D), which suggests strong association to the structure. These results indicate the possibility that tubulin is involved in the construction of an essential cytoskeletal element (or backbone) in the PSDL structure, either alone or in association with other MEC proteins.

The isolated PSDL structure may not be an artifact formed after the treatment of synapses with detergents, considering that the structure in which PSDL structures with abundant tubulin were buried was obtained from SPM after mild detergent treatment. Such

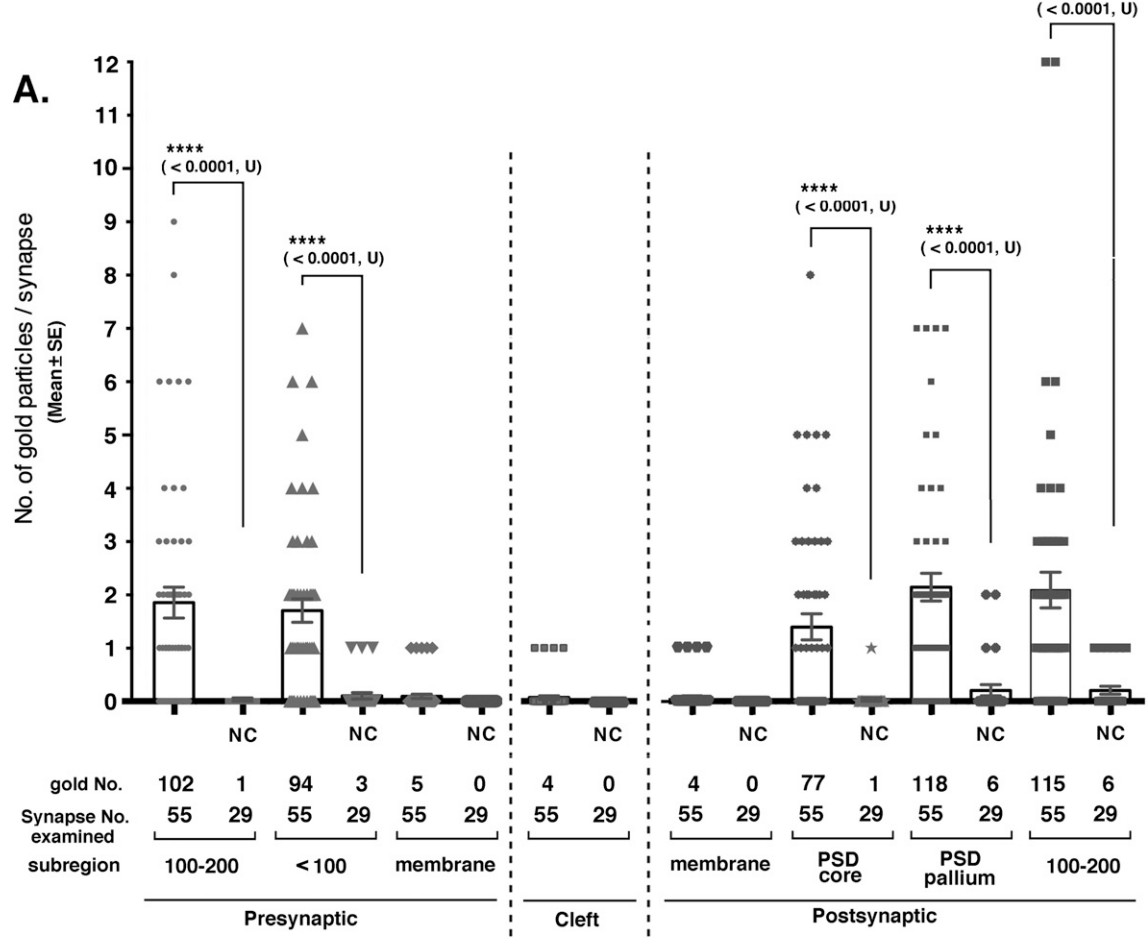

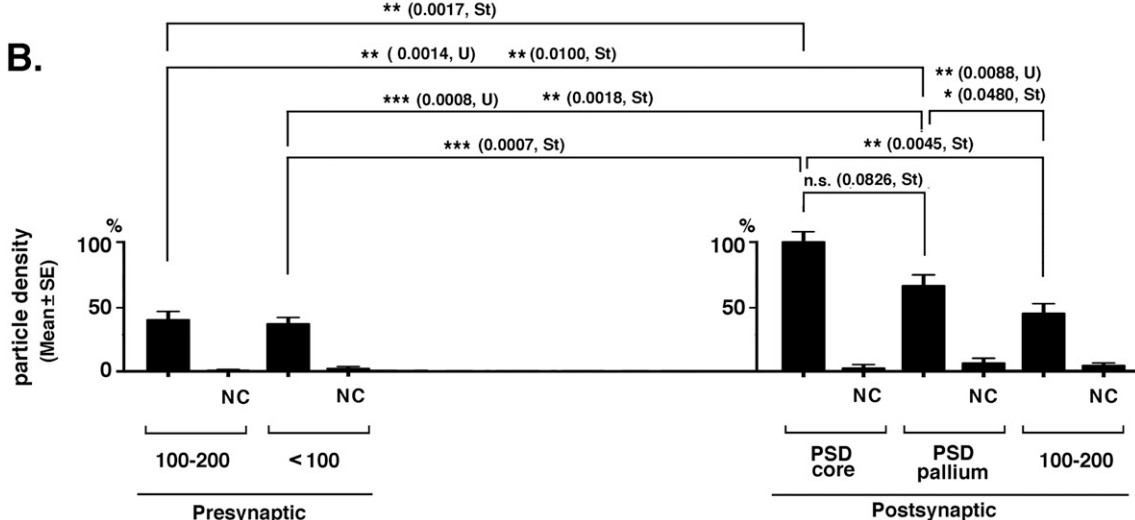

**Figure 8. Quantitative analysis of tubulin immunoreactivity in the synaptic areas of mouse cerebral cortex.**
The distribution of tubulin immunoreactivity was investigated in the mouse cerebral cortex by post-embedding immunogold labeling EM. **(A)** Number of gold particles (mean ± SE) in each synaptic region. Significance was examined by Mann–Whitney U test (U) because gold particles were not normally distributed in all regions. *P*-values are indicated in parentheses. Samples that were not significantly different from the negative controls were not indicated. **(B)** Density of the immunogold particles in each synaptic region. For the relative density of the immunogold particles, real counts were divided by the thickness of the PSD core, PSD pallium, and presynaptic <100 nm and 100–200 nm regions (30, 70, 100, and 100 [nm], respectively), and the density in the PSD core region was set at 100%. Differences judged to be significant by Mann–Whitney U tests (U) and *t* test (St) are shown in the graph. Nonsignificant cases, except for one case, are not indicated. NC, negative control using preabsorbed antibody.
Source data are available for this figure.

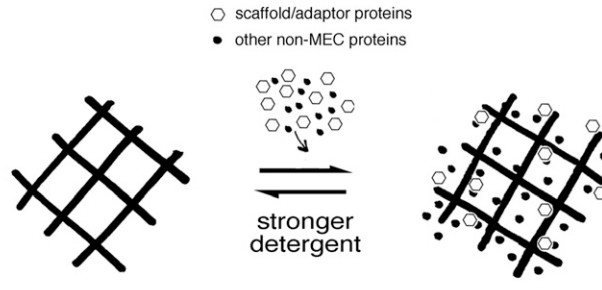

○ scaffold/adaptor proteins
● other non-MEC proteins

stronger
detergent

backbone structure
of PSD lattice

"lean" ~ "enriched"
PSD lattice

**Figure 9.   Relationship between PSDL backbone structure and "lean" and "enriched" type PSDL.**
The PSDL backbone functions as a platform from which PSD can grow by associating with various nonminimum essential cytoskeleton proteins, other cytoskeletons, and other cellular components. Treatment of synaptic plasma membrane with a relatively stronger detergent dissociated most nonminimum essential cytoskeleton proteins. Note that the backbone structure does not contain the typical PSD scaffold/adaptor proteins, CaMKII, and actin. Therefore, it is different from the three-dimensional assembly constructed by postsynaptic scaffold/adaptor proteins.

intermediate structures should not appear if the PSDL is newly formed after the destruction of synapses by the detergent. This result indicates that a PSDL with an abundance of tubulin exists in vivo before detergent treatment. Further approaches might be necessary to fully rule out the possibility that preparation artifacts may contribute to the observed pattern, however, it is highly plausible under the present conditions that tubulin is an integral component of the PSDL and a key molecule that forms the structure. This conclusion is in good agreement with prior biochemical studies, which suggested that tubulin localizes inside the PSD (Ratner & Mahler, 1983; Yun-Hong et al, 2011). It is currently completely unknown how non-MT tubulin plays a structural role in the PSDL.

Tubulin associated with the purified PSDL structure is not in the MT form because no MT was identified in the purified PSDL. Tubulin present in the in situ PSD and PSDL may also be in a non-MT form because MTs are not observed in the in situ PSD, although MTs run alongside PSD and are connected with PSD in some cases (Westrum et al, 1980). The presence of non-MT tubulin (either monomeric, dimeric, or polymeric) has been reported in SPM (Matus et al, 1975), and is referred to as membrane tubulin or integral tubulin (Zisapel et al, 1980; Babitch, 1981; Strocchi et al, 1981; Wolff, 2009). It has been suggested that the hydrophobic tubulin in the membrane is linked to protein complexes, such as certain types of cytoskeletons (Sonesson et al, 1997), and tubulin present in the PSD may also be in non-MT form (either monomeric, dimeric, or polymeric) (Matus et al, 1975). However, an actual image of the structure containing non-MT or membrane tubulin in SPM and PSD has yet to be obtained. The conoid in Toxoplasma gondii is currently the only confirmed physiological structure consisting of non-MT tubulin (Hu et al, 2002). Tubulin in the PSDL may also be non-MT tubulin.

Tubulin associated with PSDs has been reported but not verified. Tubulin has been identified in all purified PSDs and the PSD-95 protein complex (Dosemeci et al, 2007; Fernandez et al, 2009). The presence of tubulin in the brain PSD was first demonstrated immunohistochemically at the EM level using antiserum to tubulin

(Matus et al, 1975), and then by monoclonal antibodies specific for β-tubulin (Caceres et al, 1984); the immunoreactivity was determined by 3,3'-diaminobenzidine in both cases. Importantly, tubulin immunoreactivity was confined to the PSD and did not expand to the spine head cytoplasm under the condition where MAP2-immunoreactivities filled the whole spine heads (Caceres et al, 1984). The report suggests a higher content of tubulin in PSDs than in the spine head cytoplasm. Immunogold labeling of tubulin in the brain PSD has not been reported for a long time until recently (Wu et al, 2020). The presence of tubulin in the PSD has been supported by the immunofluorescence detection of tubulin in spine heads (Cáceres et al, 1986; Huang et al, 2006; Hu et al, 2008; Cheng et al, 2009a). Expression experiments using GFP- or YFP-tubulin also demonstrated fluorescent signals in spine heads of neuronal dendrites (Gu et al, 2008; Hu et al, 2008; Cheng et al, 2009a, 2009b; Merriam et al, 2013), although signals in the spine heads were weak and the potential for overexpression effects was not ruled out.

Overall, the presence of tubulin in the in situ PSD seems to have not yet been widely accepted as a consensus possibly because there had been no report on the detailed distribution of tubulin in the synaptic region of the brain using immunogold labeling. This might be owing to the sensitive nature of tubulin to nearby conditions. For example, brief cold treatment of brain tissues causes the exodus of some PSD proteins, including tubulin from PSD (Fiala et al, 2003; Cheng et al, 2009a), and artificially reduces tubulin content in the PSD. Thus, cold treatment of the brain, even for a short time, causes the lack of tubulin detection in the in situ PSD. Other conditions, although not clearly known, may also affect the detection of tubulin in the in situ PSD. Similar instability is also observed in MTs in spine head, in which MT appears only transiently. This extremely dynamic nature reduced the opportunity of detection of MT in the spine heads by EM (Hu et al, 2008).

Despite these difficulties, our immunogold labeling protocol detected tubulin immunoreactivity in and around the PSD in the brain (Figs 7 and 8). Multiple tubulin molecules are distributed in a single PSD core, such as immediately underneath the SPM, at the cytoplasmic end and central portion of the PSD core. Immunogold was also localized on the SPM, which suggests the insertion of tubulin molecules into the membrane, and anchorage of tubulin-related structures to the membrane. Tubulin signals were also distributed in the synaptic cleft. It is unclear whether these signals indicate extracellularly localized tubulin or membrane-inserted tubulin exposed to the synaptic cleft, or nonspecific signals. The absence of a report on tubulin in the synaptic cleft may support either of the latter two cases. Tubulin signals were also present in the assumed PSD pallium region (previous name: subsynaptic web) (Figs 7 and 8), a region extending from the PSD core region (30 nm deep from the postsynaptic membrane) to the 50-nm deep cytoplasmic region (Dosemeci et al, 2016). Although this region is not constantly highly electron-dense, it is a site for the translocation of some PSD proteins, and becomes highly electron-dense when PSD proteins, in particular, CaMKII, are translocated into the region (Tao-Cheng, 2019). Our study demonstrated that tubulin was distributed in both the PSD core and pallium region, which suggests that the PSDL structure underlies both the PSD core and pallium regions.

In addition to in the PSD, tubulin signals were widely detected in some synaptic terminals and dendritic spine heads (Fig 7). The difference or similarity between the PSDL and the widespread region in presynaptic terminals and dendritic spines is unclear at present. Cytoskeletal meshwork is also identified at a distance from the PSD in the dendritic spine of the high-pressure frozen brain tissue (Rostaing et al, 2006); this structure appears to be an actin-based meshwork. The association of tubulin with this actin-based spine meshwork has not yet been examined. In the presynaptic terminal, a widely spread matrix structure has been identified (Siksou et al, 2007). This presynaptic matrix is different from the PSDL structure because the structure is composed of synaptic vesicles tethered by short fibers. Presynaptic tubulin, and post-synaptic tubulin at a distance from the PSD in the spine heads might be related to the MTs observed in these regions (Gray et al, 1982; Fiala et al, 2003), which, in particular in the spine head, appear transiently but repeatedly (Hu et al, 2008). These points should wait for future studies.

Actin may not contribute to the formation of the backbone structure of the PSDL because its content in the isolated PSDL structure is extremely low (Fig 6B), despite being grouped as an MEC. The nonparticipation of actin to the PSD backbone structure is supported by the finding that little actin immunoreactivity (immunogold) was detected within the PSD and that this was localized in the spine head cytoplasm (Rostaing et al, 2006). The surface localization of actin on the purified PSD (Ratner & Mahler, 1983) also supports our conclusion. $\alpha$-IN may also not be an essential component for constructing PSDL structures. The higher content of actin and $\alpha$-IN in the initial PSDLs than in the new PSDLs may be because proteins, such as actin and $\alpha$-IN, which are not closely linked to in situ PSDLs, are contaminated in the initial PSDL preparation (Fig 5A and B). It is interesting that, among various PSD scaffold proteins, only PSD-95 was categorized as a MEC based on proteomics analyses; however, its concentration was decreased in the lean-type PSDL, like other scaffold/adaptor proteins. PSD-95 may have a specialized role in the construction of the PSDL. Alternatively, this may be simply owing to the abundance of PSD-95 compared with other scaffold/adaptor proteins.

## Non-MEC proteins and their possible roles for PSDL

In addition to the MEC components, multiple proteomics analyses identified non-MEC components of the PSDL (Fig 5). The non-MEC proteins are associated with the MEC-based backbone structure of the PSDL and are easily solubilized with high concentrations of OG. Thus, they are not essential structural components necessary for constructing the backbone structure for the PSDL.

Non-MEC components of PSDL (0.75% OG) contain various proteins (Table S4). They can be functionally categorized into two groups: proteins related to the structure and those related to nonstructural functions. The former includes scaffold/adaptor proteins containing the Dlg family, cell adhesion molecules, junctional proteins, and cytoskeletal proteins. Some of them may be related to the formation of scaffold/adaptor protein assembly and nanocolumn domain (or nanomodule) structures in the PSD. The latter includes receptor proteins, channels/transporter proteins, regulators for cytoskeletal dynamics, and proteins related to other cellular signaling.

Various kinds of non-MEC scaffold/adaptor proteins may add additional structural elements to the PSDL backbone. In other words, the PSDL backbone structure may work as a supporting platform to which three-dimensional structures woven by scaffold/adaptor proteins are anchored (Fig 9). PSD scaffold/adaptor proteins, including PSD-95, with synaptic cell adhesion molecules and neurotransmitter receptors, may lead to the formation of trans-synaptic nanodomain column structures (Tang et al, 2016). Structure-related non-MEC proteins may also play a role in tethering nascent PSD with other cytoskeletons and cellular compartments surrounding PSD. Actin-related, MT-related, and spectrin-related cytoskeletons may also be linked to the PSDL backbone structure. Many regulatory proteins for these cytoskeletons are non-MEC components (Table S4). The relationship between the PSDL backbone and membrane trafficking, cell fusion machinery, and the proteasome is also plausible.

The results presented in this study suggest that various functional and structural subordinate proteins are associated over the PSD backbone structure to make the enriched PSDL structure, which may be an intermediate structure leading to the formation of mature PSD (Fig 9). The non-MT tubulin-based PSDL is formed in the early stage of synaptogenesis (Fig 2B-c and B-d and 3A-c), which supports the idea that it functions as a platform from which PSD can grow.

The content of fodrin, PSD-95, GKAP, CaMKII$\alpha$, and CaMKII$\beta$, but not tubulin or ATP synthase, tended to decrease in the PSDL preparations as the OG concentration increased (Fig 4D). This tendency was confirmed by immunogold EM observation, where GKAP and Homer1 were more enriched than Shank1 in the enriched-type PSDL but not in the lean type (Fig 6B). Together with the deviated abundance of immunogold particles for PSD-95, Homer1, CaMKII$\alpha$/$\beta$, these results suggest that these proteins are sub-ordinate PSDL-associated proteins. These associating non-MEC proteins were localized on the additional non-membranous structures to the enriched PSDLs (Fig 6C). In other words, the association between the backbone structure of the PSDL and the non-MEC proteins may occur via protein–protein interactions, but not indirect interactions through the membrane. These results suggest that non-MEC proteins form large protein complexes by associating with the backbone structure of the PSDL, the accumulation of which leads to the maturation of the PSD.

It is interesting that there was a difference in the content in the PSDL structure among the scaffold/adaptor PSD proteins examined. PSD-95 = Homer1 > GKAP > Shank1 on the PSDL (0.75% OG) structure. Considering the PSD scaffold/adaptor assembly model, which consists of PSD95–GKAP–Shank protein webs (Feng & Zhang, 2009), these differences in concentration suggest the location of the PSDL in the PSD core region. In other words, the PSDL structure is localized close to the postsynaptic membrane, where PSD-95 is localized, but may not be expanded deeply into the cytoplasmic region, where the Shank layers are localized. The idea agrees well with the association of PSDL with Homer1 and GKAP, which are localized in between PSD-95 and shank layers (Tao-Chen, 2014, 2015). Shank is localized in the PSD pallium region and spine cytoplasm (Tao-Cheng, 2010, 2019), and our immunogold histochemistry suggested that the PSDL structure is an underground structure in both PSD core and pallium. However, Shank amount is

very low in the purified PSDL. One explanation for this apparent discrepancy is that the purified PSDL is a nascent structure, which matures by making cross-links with Shank webs, which develop separately from the PSD-95-GKAP webs (Li et al, 2017). Alternatively, most of Shank proteins were not translocated into the PSD pallium region in the brains used. Further studies will be needed to confirm this.

Proteomics analyses identified unique categories of proteins in the purified PSDLs, including inhibitory synapse proteins such as gephyrin and GABA receptors, glial components, presynaptic components, mitochondrial components, nuclear-related proteins, and translation-related components. Simple questions are why both excitatory and inhibitory components are co-purified and whether glial components, presynaptic proteins, and nuclear proteins are associated with the PSDL in vivo or contamination. For certain proteins, explanation can be provided. For example, some presynaptic components are tethered to connect pre- and post-synaptic structures. Glial fibrillary acidic protein might be tethered to the synapse structure in a tripartite synapse structure. Excitatory and inhibitory scaffolds might be associated with each other in spines dually innervated with excitatory and inhibitory synaptic terminals (Villa et al, 2016). However, it is not possible to exclude the possibility of contamination of the PSDL fraction. This problem cannot be avoided in biochemically purified structures (Suzuki et al, 2007, 2011). These proteins are also present in purified PSDs (Jordan et al, 2004; Yoshimura et al, 2004; Suzuki et al, 2011). This might be owing, at least in part, to the fact that sensitive MS detects trace amounts of contaminating proteins. This problem can be solved by a one-by-one examination using other localization analyses, such as the immunohistochemical approach. We added comments on the mitochondrial proteins in the next section.

### Mitochondria-related components in the PSDL preparation

The new PSDL preparation purified in this study contained a large number of mitochondria-related proteins, although their contents, except for ATP synthase and ANT, appear to be low. Tubulin, the most abundant protein in the PSDL, can interact with mitochondria and is an inherent component of mitochondria (Carre et al, 2002; Guzun et al, 2011). Therefore, whether the PSDL structure is derived from mitochondria is a critical question. However, the present study and the current literature do not support this possibility. The morphology and profile of the protein components of the mitochondria-derived detergent-insoluble material (Ko et al, 2003) are different from our PSDL structures. At the same time, SDS–PAGE profile (Fig 3A-a) and morphology of our PSDL are similar to those of PSD (Figs 2B and S3) (Suzuki et al, 1984, 2018). Therefore, it is unlikely that the purified structure is derived from mitochondria.

The second question is with regard to where the mitochondria-related proteins come from. ANT, ATP synthase, and hexokinase form a protein complex together with voltage-dependent anion-selective channel protein (VDAC) in the mitochondrial membrane. This complex is maintained after treatment with detergent (Wagner et al, 2001; Ko et al, 2003; Guzun et al, 2011). All the components of this complex were present in the PSDL preparation. Such protein complexes, as well as other mitochondria-residing proteins, could be released from mitochondria after

treatment with detergent and co-sediment with the PSDL structure. Another possibility is that mitochondria-related proteins are derived from the plasma membrane, outside the mitochondria. Previous studies have reported that several mitochondria-residing proteins are also localized outside mitochondria (Zhao et al, 2004; Angrand et al, 2006; Yano et al, 2006; Yonally & Capaldi, 2006; Detke & Elsabrouty, 2008; Mishra et al, 2010; Loers et al, 2012) (Table S5).

Finally, ANT protein in the PSDL preparation was detected in SDS–PAGE (Fig 3A) and emPAI estimation calculated its abundance in the preparation (Fig S2). However, ANT content in the PSDL preparation fluctuated among preparations, as seen in Figs 3B-a and 4C, and immunogold EM observation showed that a very little amount of ANT, if any, was distributed on our PSDL structure (Fig 6B). Therefore, it is unlikely that ANT plays a major role in the construction of PSDL structure. ATP synthase was not abundant on the PSDL structure compared with tubulin (Fig 6B), and its content was extremely low in the purified PSD, compared with tubulin and PSD-95 (Fig 4D–F). However, it has been reported that ATP-synthase, possibly localized on the plasma membrane, interacts with tubulin and PSD-95 (Dosemeci et al, 2007; Fernandez et al, 2009; Uezu et al, 2016), suggesting relation with the PSD or PSDL. It should wait for future studies how such non-mitochondria-localized proteins are related to the PSD or PSDL.

### Conclusions

The components of the PSDL were identified using a new PSDL preparation method and categorized as either MEC or non-MEC proteins. Our results suggest that non-MT tubulin is related to the backbone structure of the PSDL, with various functional or structurally subordinate proteins associated with the PSDL structure. The PSDL structure may underlie both PSD core and pallium region. Non-MT tubulin PSDL structures may play an important role as a platform to which PSD scaffold/adaptor proteins and various PSD-functioning molecules become associated, while synapses mature and reorganize.

# Materials and Methods

### Materials

The chemicals and antibodies used in this study are listed in Tables 2–4. All chemicals unlisted in Tables 2–4 are of reagent grade. Tubulin was purified from porcine brains by three cycles of polymerization and depolymerization according to the method of Shelanski et al as modified by Ihara et al, and stored at –70°C (Suzuki et al, 1986).

### Ethical approval/animals

Animals were handled in accordance with the Regulations for Animal Experimentation of Shinshu University. The animal protocol, together with animal handling, was approved by the Committee for Animal Experiments of Shinshu University (approval no. 240066).

**Table 2. List of major chemicals used in this study.**

| Chemicals (abbreviated names) | Code No. | RRIDs | Providers |
|---|---|---|---|
| Butorphanol | 42408-82-2 | Not found | Meiji Seika |
| Colloidal gold total protein stain | 1706527 | Not found | Bio-Rad Raboratories, Inc. |
| Iodoacetamide (IAA) | 095-02891 | Not found | WAKO Pure Chemical Industries. Ltd. |
| Immobilon-P | IPVH00010 | Not found | Millipore Corporation |
| ImmunoStar LD | 292-69903 | Not found | WAKO Pure Chemical Industries. Ltd. |
| IR-Gold | 17412 | Not found | Polysciences |
| Medetomidine | CS-0734 | Not found | Fujita Pharmaceutical Company |
| Midazolam | 59467-70-8 | Not found | Sandoz |
| MPEX PTS reagent | 5010-21360, 21361 | Not found | GL Sciences Inc. |
| Nano-W | 2018 | Not found | Molecular probes |
| n-Octyl-β-D-glucoside (OG) | 346-05033 | Not found | Dojindo Laboratories |
| Oriole | 161-0495 | Not found | Bio-Rad |
| Protease inhibitor cocktail | P8340 | Not found | Sigma-Aldrich |
| Silver staining kit | AE-1360 | Not found | ATTO (Atto Bioscience & Biotechnology) |
| SYPRO ruby protein gel stain | 505654 | Not found | Lonza Rockland, Inc. |
| Triton X-100 (TX-100) | 581-81705 | Not found | WAKO Pure Chemical Industries. Ltd. |

Based on the national regulations and guidelines, all experimental procedures were reviewed by the Committee for Animal Experiments and finally approved by the president of Shinshu University.

Wistar rats (male, 6 wk old, body weight: 150 ± 8 g, specific pathogen–free), pregnant rats (body weight: 220–250 g) (Slc: Wistar [SPF], RRID: RGD_2314928), and C57BL/6J mice (male, 4–5 mo old, body weight: 20–30 g, specific pathogen–free) (C57BL/6JJmsSlc, RRID: IMSR_JAX:000664) were purchased from Japan SLC, Inc.. The brains of 6-wk-old rats were collected on the day of delivery from the company. Pregnant and newborn rats were housed at 23 ± 3°C at a constant humidity under a 12-h light/dark cycle in a flat-floor cage made of resin (polysulfone) with paper chips. Rats were provided with free access to tap water and standard rat chow. Newborn rats were grown until the seventh day after birth (body weight: 15 ± 1.2 g and 14.5 ± 0.8 g for males and females, specific pathogen–free), at which point their brains were collected.

**Preparation of SPM and conventional PSD**

SPMs were prepared from Wistar rats (see previous section) either in the presence or absence of the antioxidant reagent IAA, as described previously (Suzuki et al, 2018, 2019). Purified SPMs were stored unfrozen in buffers containing 50% glycerol at −30°C. Storing for longer at −80°C did not cause any substantial deterioration of the samples.

"Conventional" TX-PSD was prepared from the forebrain of 6-wk-old rats by treatment with 0.5% TX-100 of synaptosomes or SPM (short or long procedure, respectively) (Cohen et al, 1977; Suzuki et al, 2019). "Conventional" n-Octyl-β-D-glucoside (OG)-PSD was prepared from the forebrain of 6-wk-old rats, following a short procedure for "conventional" TX-PSD purification, with OG (1%) instead of TX-100 (0.5%). We adopted 1% OG based on our previous

study, by which recovery of OG-12, another type of OG-insoluble PSD was higher than the 0.5%OG (Zhao et al, 2014). Conventional PSDs were prepared in the absence of IAA following the original protocol (Cohen et al, 1977). In the case of conventional OG-PSDs, two PSD fractions (OG-PSD [light] and OG-PSD [heavy]), were obtained at and below the 1.5–2.1 M sucrose interface, respectively. Conventional OG-PSDs retrieved immediately after discontinuous SDG ultracentrifugation were not treated with OG/KCl, unlike conventional TX-PSD purification. OG-PSD (heavy) was washed once by adding 2× H$_2$O, followed by centrifugation at 15,780g for 20 min. The resulting pellet was resuspended in 400 μl of 5 mM Hepes/KOH (pH 7.4) containing 50% glycerol and hand-homogenized. OG-PSD (light) was not washed because of the small amount. Both conventional OG-PSDs were stored at −30°C or at −80°C for a longer storage.

**Purification of PSDL and 1% OG-12 (one type of OG-insoluble PSD preparation)**

PSDL preparations were purified from Wistar rats (the same species used for SPM and PSD purification) following our published protocol (Suzuki et al, 2019) (Fig 1B). We adopted 1% OG for purification of standard PSDL preparations based on our previous study (Zhao et al, 2014; Suzuki et al, 2018). The concentration of OG was changed as required. Briefly, SPM (3-mg protein) prepared from the forebrains of 6-wk-old or 7-d-old rats in the presence of IAA, if not stated otherwise, was treated with 1% OG for 30 min at 4°C in 10.5 ml of 20 mM Tris–HCl buffer (pH 7.4) containing 150 mM NaCl and 1 mM EDTA (TNE buffer) supplemented with protein inhibitor mixtures (p8340; Sigma-Aldrich) at 1/200 dilution and IAA (2 mM) (detergent: protein ratio, 35:1 [w/w] at 1% detergent). The detergent-treated solution was mixed with an equal volume of TNE buffers containing

**Table 3. List of antibodies used for immunogold EM.**

| Antibodies used for immunogold labeling | RRID (−), not found or no exact match | Catalog or clone no. | Company | Mono or poly | Animals for antibody production | Dilution used |
|---|---|---|---|---|---|---|
| Anti-tubuin | Not registered | [a] | Produced by Dr. Fujii, Shinshu University | Polyclonal | Rabbit | 1/20 |
| Anti-$\alpha$-tubulin | (−) | RB9281-P0 | Thermo Fisher Scientific K.K. | Polyclonal | Rabbit | 1/20 |
| Anti-$\beta$-tubulin | AB_609915 | T-5201 | Sigma-Aldrich | Monoclonal | Mouse | 1/20 |
| Anti-$\beta$III-tubulin | (−) | MMS-435P | Covance | Monoclonal | Mouse | 1/20 |
| Anti-$\beta$-actin[b] | AB_626632 | sc-47778 | Santa Cruz Biotechnology Inc. | Monoclonal | Mouse | 1/20 |
| Anti-$\alpha$-internexin | AB_91800 | AB5354 | Chemicon International | Polyclonal | Rabbit | 1/500[c] |
| Anti-spectrin (nonerhtyroid)[d] | AB_11214057 | MAB1622 | Chemicon International | Monoclonal | Mouse | 1/20 |
| Anti-CaMKII$\alpha$ | (−) | 6G9 | Chemicon International | Monoclonal | Mouse | 1/20 |
| Anti-CaMKII$\beta$ | (−) | 3232SA | Gibco BRL | Monoclonal | Mouse | 1/20 |
| Anti-PSD-95 | (−) | MA1-045 | ABR | Monoclonal | Mouse | 1/20 |
| Anti-Homer 1 | AB_1950505 | GTX103278 | GeneTex, Inc. | Polyclonal | Rabbit | 1/20 |
| Anti-shank1 | AB_2270283 | N22/21 | UC Davls/NIH NeuroMab facility | Monoclonal | Mouse | 1/20 |
| Anti-GKAP (Pan-SAPAP) | ABJ0671947 | N127/31 | UC Davls/NIH NeuroMab facility | Monoclonal | Mouse | 1/20 |
| Anti-ATP5A1, C-term | AB_10618791 | GTX101741 | GeneTex, Inc. | Polyclonal | Rabbit | 1/20 |
| Anti-ANT | AB_671086 | sc-9299 | Santa Cruz Biotechnology Inc. | Polyclonal | Goat | 1/20 |
| Anti-Mouse IgG (H+L)-gold label | (−) | EMGMHL5, EMGMHL10 | BBI solutions | Polyclonal | Goat | 1/50 |
| Anti-Rabbit IgG (H+L)-gold label | (−) | EMGAR5, EMGAR10 | BBI solutions | Polyclonal | Goat | 1/50 |
| Anti-Goat IgG (H+L)-gold label | (−) | EMRAG10 | BBI solutions | Polyclonal | Rabbit | 1/50 |

[a]Anti-tubulin antibody was produced in rabbit using pig tubulin as antigen, and affinity-purified (Liu et al, 2013).
[b]Anti-$\beta$-actin (sc-47778) was used because anti-pan-actin antibody (pan Ab-5; Thermo Fisher Scientific, UK, RRID, AB_10983629) did not label well the 1% OG-11B and PSD.
[c]This dilution is due to the stock solution which is 1/100 diluent of the original solution.
[d]Anti-fodrin.

80% sucrose, overlaid with TNE buffers containing 30% sucrose, and then 5% sucrose (3.5 ml each/tube) in six centrifuge tubes, and centrifuged (256,000$g_{max}$, 30 h, 4°C). The positions of 11 fractions (955 $\mu$l each) were marked on each centrifuge tube and numbered from the top. The solution contained in fractions 1–10 was discarded. The upper portion (825–755 $\mu$l) of fraction 11 (1% OG-11U) and the bottom portion (1% OG-11B) (130–200 $\mu$l) were collected. 1% OG-11U and 1% OG-11B were diluted 3.5- to 5-fold with 5 mM Hepes/KOH (pH 7.4), and the insoluble (IS) components of these fractions were pelleted by centrifugation (100,000$g_{max}$, 30 min, 4°C). These pellets were resuspended in 1 ml of 5 mM Hepes/KOH (pH 7.4), and ultracentrifuged again. The final pellets were resuspended by repeatedly pipetting in 100 $\mu$l of 5 mM Hepes/KOH (pH 7.4) containing 50% glycerol. The resuspended pellet was hardly visible to the naked eye after the last two ultracentrifugation steps. The final solutions were neither hand-homogenized nor vortexed to avoid any loss or protein denaturation.

1% OG-12, fraction 12 prepared from SPM treated with OG (another type of PSD preparation) is a pellet obtained after the SDG ultracentrifugation during purification of PSDL (1% OG) (see Fig 1B). OG-12 was prepared in the presence or absence of 2 mM IAA from 500 $\mu$g of SPM protein and finally suspended in 150 $\mu$l of 5 mM Hepes/KOH (pH 7.4) containing 50% glycerol. These preparations were stored at −30°C or −80°C for longer storage.

### Immunogold negative-staining EM and subsequent analyses

Negative staining coupled with the immunogold technique using 10-nm gold particles was carried out as described previously (Suzuki et al, 2018). The dilutions of the primary antibodies were based on a previous study (Swulius et al, 2010), and those of the second antibodies were carried out according to the manufacturer's instructions. The specimens were examined under a JEOL JEM-1400EX EM (JEOL) at 80 kV, and images were taken using a 4,008

**Table 4. List of antibodies used for Western blotting.**

| Antibodies used for Western blotting | RRID (−), not found or no exact match | Catalog or clone No. | Company | Mono or poly | Animals for antibody production | Dilution used |
|---|---|---|---|---|---|---|
| Anti-tubuin | Not registered | [a] | Produced by Dr. Fujii, Shinshu University | Polyclonal | Rabbit | 1/20,000–1/50,000 |
| Anti-$\alpha$-tubulin | (−) | RB9281-P0 | Thermo Fisher Scientific K.K. | Polyclonal | Rabbit | 1/5,000 |
| Anti-$\beta$-tubulin | AB_609915 | T-5201 | Sigma-Aldrich | Monoclonal | Mouse | 1/1,000 |
| Anti-$\beta$III-tubulin | (−) | MMS-435P | Covance | Monoclonal | Mouse | 1/1,000 |
| Anti-$\beta$-actin | AB_626632 | sc-47778 | Santa Cruz Biotechnology Inc. | Monoclonal | Mouse | 1/5,000 |
| Anti-spectrin (nonerhtyroid) [b] | AB_11214057 | MAB1622 | Chemicon International | Monoclonal | Mouse | 1/250 |
| Anti-PSD-95 | (−) | 610495 | BD Transduction Laboratories | Monoclonal | Mouse | 1/1,000 |
| Anti-$\alpha$-internexin | AB_91800 | AB5354 | Chemicon International | Polyclonal | Rabbit | 1/50,000 |
| Anti-CaMKII$\alpha$ | (−) | 6G9 | Chemicon International | Monoclonal | Mouse | 1/20,000 |
| Anti-CaMKII$\beta$ | (−) | 3232SA | Gibco BRL | Monoclonal | Mouse | 1/20,000 |
| Anti-Homer 1 | AB_1950505 | GTX103278 | GeneTex, Inc. | Polyclonal | Rabbit | 1/20,000 |
| Anti-shank1 | AB_2270283 | N22/21 | UC Davls/NIH NeuroMab facility | Monoclonal | Mouse | 1/200 |
| Anti-GKAP (Pan-SAPAP) | ABJ0671947 | N127/31 | UC Davls/NIH NeuroMab facility | Monoclonal | Mouse | 1/250 |
| Anti-ATP5A1, C-term | AB_10618791 | GTX101741 | GeneTex, Inc. | Polyclonal | Rabbit | 1/3,000 |
| Anti-ANT | AB_671086 | sc-9299 | Santa Cruz Biotechnology Inc. | Polyclonal | Goat | 1/1,000–1/3,000 |
| Anti-Mouse IgG-HRPO[c] | AB_772210 | NA931 | GE Healthcare | Polyclonal | Sheep | 1/5,000 |
| Anti-Rabbit IgG (H+L)-HRPO[c] | AB_10682917/AB_437787/AB_437787 | 401315 | Millipore (purchased from Calbiochem) | Polyclonal | Goat | 1/20,000–1/50,000 |
| Anti-Goat IgG-HRPO[c] | AB_11214432/AB_92420 | AP107P | Chemicon International | Polyclonal | Rabbit | 1/20,000 |

[a]Anti-tubulin antibody was produced in rabbit using pig tubulin as antigen, and affinity-purified (Liu et al, 2013).
[b]Anti-fodrin.
[c]HRPO, horseradish peroxidase.

× 2,672-pixel elements CCD camera (Gatan SC1000; Gatan Inc.). The contrast of the images was edited in Photoshop to make the gold particles clearly visible. $\gamma$-Contrast was not modulated. The area of the PSDL structure was measured by Image J 1.51r (NIH). For quantitation, PSDL structures with 30,000–350,000 $nm^2$ (corresponding to 195–668 nm in diameter if they are supposed to be circles) were randomly selected and the gold particles were counted.

## Post-embedding immunogold labeling EM and subsequent analyses

Mice were anesthetized with a mixture of butorphanol (Meiji Seika), medetomidine (Fujita Pharmaceutical Company), and midazolam (Sandoz), and perfused with physiological saline (20 s at 20 ml/60 s) and 2% paraformaldehyde and 0.25% glutaraldehyde in phosphate buffer (80 s at 20 ml/60 s) through the heart at room temperature. Central parts (1 mm wide, 1 mm deep) of 1-mm thick coronal sections at the Bregma in the cerebral cortex (both hemispheres) were dissected. The tissue blocks from the right hemisphere (1 × 1 × 1 mm) were immersed in the same fixative for 3 h at room temperature, dehydrated in a gradient series of ethanol at –15°C, and embedded in acrylic resin, LR-Gold (Polysciences). The resin was polymerized under an ultraviolet beam at –15°C for 24 h. Tissues were not fixed with osmium to maximally retain antigenicity. Ultrathin sections (~100 nm) were placed on nickel grids covered with the formvar membrane. The sections were blocked with 10% goat serum for 30 min, incubated with anti-tubulin antibody overnight at 4°C, and subsequently incubated with goat anti-rabbit IgG antibody conjugated to 10-nm colloidal gold (BBI Solutions) at room temperature for 30 min. After post-fixation with 1% glutaraldehyde in $H_2O$ for 10 min, the specimens were counterstained with 1% uranyl acetate for 10 min and examined under a JEOL JEM-1400Flash (JEOL) at 80 kV. Images were captured using an sCMOS camera (EM-14661FLASH) and the contrast of the images was edited using Photoshop.

The distribution of tubulin immunoreactivity was investigated in the mouse cerebral cortex by post-embedding immunogold labeling EM. For quantitation, asymmetric synapses with PSDs ranging from 138 to 522 nm in length were randomly selected and the

number of immunogold particles was counted in the following synaptic subregions: pre- and post-SPMs, synaptic cleft, PSD core (strongly electron-dense portion of PSD immediately below the postsynaptic membrane [Dosemeci et al, 2016]), assumed PSD pallium region (region extending from the PSD core but within 100 nm from the postsynaptic membrane) (Dosemeci et al, 2016), postsynaptic regions located between 100 and 200 nm from the postsynaptic membrane, and presynaptic regions located <100 nm and between 100 and 200 nm from the presynaptic membrane. Type II synapses were not counted. Specimens processed without primary antibodies or incubated with preabsorbed anti-tubulin antibody were used as NCs. Preabsorption of anti-tubulin antibody was carried out by incubating solution containing anti-tubulin antibody with Immobilon-P (Millipore) to which purified tubulin was electroblotted after SDS–PAGE.

## Electrophoresis and Western blotting

SDS–PAGE was carried out using 10% polyacrylamide gel, unless stated otherwise. The gels were stained with SYPRO Ruby or Oriole. The fluorescent signals were captured using a WSE-5200 Printgraph 2M (ATTO Bioscience & Technology). The amount of protein in the preparations was estimated based on the densitometry of the SYPRO Ruby signals of total proteins separated on the electrophoretic gel using the standard samples. The protein concentration was determined based on bovine serum albumin. Western blotting was carried out using ImmunoStar chemiluminescent substrate (Wako Pure Chemical) and visualized with a CCD video camera system (myECL; Thermo Fisher Scientific Inc.). The amount of protein in the immuno-blot bands was normalized to tubulin contained in the same lane. Contrast of images of SDS–PAGE and Western blotting was modulated with Photoshop although they are greatly dependent on the exposure conditions.

## Mass spectrometric analysis

The comprehensive identification of proteins in the PSDL preparations was carried out using the shotgun method by Shimadzu Techno Research.

The flow of shotgun proteomics is schematically summarized in Fig S1A. An aliquot of sample (20 $\mu$l) was mixed with 80 $\mu$l of ultrapure water. Then, 400 $\mu$l of methanol and 100 $\mu$l of chloroform were sequentially added to the sample to precipitate the proteins. The sample was diluted with 300 $\mu$l of ultrapure water and centrifuged at 15,000$g$ for 2 min at room temperature. The upper layer containing the OG and glycerol was removed. The lower layer, after mixing with 400 $\mu$l of methanol, was centrifuged at 20,000$g$ for 2 min, and the pellet obtained was dried by vacuum evaporation. The protein pellet was dissolved with 20 $\mu$l of MPEX PTS reagent, which contained sodium deoxycholate (NaDOC) and Sodium N-Lauroylsarconinate (SDS No. 5010-0021; GL Sciences Inc.) (Masuda et al, 2009), and centrifuged at 20,000$g$ for 2 min. The pellet was resuspended in MPEX PTS reagent. Both the soluble and insoluble materials were analyzed by MS.

Proteins reduced with DTT and alkylated with IAA were digested with trypsin at 37°C for 16 h. After removing the MPEX PTS reagent by phase transfer, the solution was concentrated to 50 $\mu$l using a vacuum concentrator. 50 $\mu$l of 5% acetonitrile solution containing 0.1% TFA were added to the residual solution. The solution was applied to a MonoSpin C18 cartridge (GL Sciences). Elution was carried out by adding 60% acetonitrile solution containing 0.1% TFA. The eluate was evaporated to dryness using a SpeedVac and the residue was reconstituted by adding 20 $\mu$l of 2% acetonitrile solution containing 0.1% FA. The protein sequences were analyzed by liquid chromatography coupled with tandem MS (LC–MS/MS) using Easy n-LC1000 (Thermo Fisher Scientific). The sample was loaded onto the chromatography column and separated with a linear gradient of mobile phase A (0.1% formic acid) and mobile phase B (acetonitrile containing 0.1% formic acid). Data acquisition was carried out using a Q Exactive PLUS mass spectrometer (Thermo Fisher Scientific). MS/MS spectra were searched using the MASCOT engine (version 2.4) (Matrix Science) embedded into Proteome Discoverer 1.4 (Thermo Fisher Scientific). The acquired MS/MS spectra were automatically searched against the SwissProt database. The related search parameters were as follows: taxonomy = rattus; enzyme = trypsin; max-missed cleavage = 1; static modifications = carbamidomethyl (C); dynamic modifications = oxidation (M); mass values = monoisotopic; peptide mass tolerance = ±10 ppm; fragment mass tolerance = ±0.02 D.

A Venn diagram was produced using Thermo Proteome Discoverer (version 1.4.1.14) (Thermo Fisher Scientific). The emPAI values (molar base), a semiquantitative measure of protein abundance based on MS data, were calculated (Ishihama et al, 2005). The proteins were manually classified, as in our previous reports (Suzuki et al, 2007, 2011). Proteins believed to be generally localized within the mitochondria were categorized into "primarily mitochondrial proteins" because many were also localized outside the mitochondria (Table S5). The names of well-known proteins are abbreviated in the list, with keratins and trypsins excluded from the list.

The main proteins were also identified by the MS analysis of the protein bands excised from polyacrylamide gels after SDS–PAGE. The protein bands stained with silver were cut, destained, reduced with dithiothreitol, alkylated with IAA, and digested in-gel with trypsin. The resulting peptide mixtures were extracted and analyzed by MS using the rat UniProtKB database and IDENTITYE, which consists of nanoACQUITY, Xevo QTof MS, and ProteinLynxTM Global SERVER (PLGS) 2.5.2 (Nihon Waters).

## Experimental design and statistical analysis

For the purification of synaptic subfractions, male rats were used to exclude sex differences, except for the preparations from 7-d-old rats, in which a mixture of males and females was used because of the difficulty of collecting male rats only for ethical reasons. The sample sizes for quantification in immunogold negative-staining EM (≥18) were based on our previous study (Suzuki et al, 2018) and a similar experiment conducted by another group (Swulius et al, 2010) but were as large as possible from the feasibility viewpoint. PSD preparations purified by various methods (Suzuki et al, 2011, 2018; Liu et al, 2013; Zhao et al, 2014) were used to strengthen the comparison data with PSD and verify the PSDL. The protein components and their amounts were analyzed using Western blotting, MS/emPAI, and immunogold negative-staining EM. The authenticity of the PSDL preparation was substantiated by repeated preparations of PSDL (0.75% OG), PSDL (1% OG), and PSDL (5% OG) (n = 5, 10, and 4, respectively) with substantially

similar protein profiles and morphologies. The specificity of the immunoreaction in the immunogold EM was verified using control specimens processed without the primary antibodies. Furthermore, tubulin immunoreactivity in the post-embedding immunogold labeling EM, anti-tubulin antibody preabsorbed with purified tubulin was also used for NCs. In Western blotting, the protein amounts were normalized to tubulin contained in the same lane to minimize variations between lanes.

Quantitative data are presented as the mean ± SE. Statistical analyses (D'Agostino-Pearson omnibus normality test, $t$ test, and Mann–Whitney's U test) were carried out using GraphPad Prism version 6.0 (GraphPad Software). Either the $t$ test or U test (both unpaired two-tailed) was used, depending on the normality of the distribution, unless stated otherwise. The results were considered statistically significant when $P < 0.05$. $P$-values and sample numbers are shown in each Figure.

## Data Availability

MS data from this publication have been deposited to jPOSTrepo (an international standard data repository for proteomes, https://repository.jpostdb.org) (Okuda et al, 2017). The accession numbers are PXD024712 for ProteomeXchange (http://www.proteomexchange.org) and JPST001106 for jPOST (Japan ProteOme STandard DataBase, https://jpostdb.org).

## Supplementary Information

## Acknowledgements

We would like to thank Editage (www.editage.jp) for their assistance with the English language editing. The authors declare no competing financial interests. The work was supported by a Grant from Institute of Medicine, Shinshu University Acdemic Assembly (to T Suzuki).

### Author Contributions

T Suzuki: conceptualization, data curation, validation, investigation, visualization, methodology, project administration, and writing—original draft, review, and editing.
N Terada: investigation.
S Higashiyama: conceptualization, methodology, and project administration.
K Kametani: investigation.
Y Shirai: investigation.
M Honda: data curation and investigation.
T Kai: data curation and investigation.
W Li: conceptualization and project administration.
K Tabuchi: conceptualization and project administration.

### Conflict of Interest Statement

The authors declare that they have no conflict of interest.

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
