## [Reviewer comments · Life Science Alliance]

Life Science Alliance

Non-microtubule tubulin-based backbone and subordinate components of postsynaptic density lattices

Tatsuo Suzuki, Nobuo Terada, Shigeki Higashiyama, Kiyokazu Kametani, Yoshinori Shirai, Mamoru Honda, Tsutomu Kai, Weidong Li, and Katsuhiko Tabuchi

DOI: <https://doi.org/10.26508/lsa.202000945>

Corresponding author(s): Tatsuo Suzuki, Shinshu University

Review Timeline:

Submission Date:	2020-10-23
Editorial Decision:	2020-12-07
Revision Received:	2021-03-15
Editorial Decision:	2021-04-09
Revision Received:	2021-04-19
Accepted:	2021-04-26

Scientific Editor: Shachi Bhatt

Transaction Report:

December 7, 2020

Re: Life Science Alliance manuscript #LSA-2020-00945-T

Prof. Tatsuo Suzuki
Shinshu University
Molecular and Cellular Physiology
3-1-1, Asahi
Matsumoto 390-8621
Japan

Dear Dr. Suzuki,

Thank you for submitting your manuscript entitled "A tubulin-based novel structure for the construction of postsynaptic density lattices" to Life Science Alliance (LSA). The manuscript was assessed by expert reviewers, whose comments are appended to this letter.

As you will note from the reviewers' comments, the main concern that has been explicitly pointed out by Reviewers 1 and 2 is that there is no data showing localization of tubulins underneath the postsynaptic membrane (in the PSD lattice structure) in situ - this could be achieved either with EM or super-resolution light microscopy. This point was also discussed in the post-review cross-commenting between the reviewers as well as discussed in detail between the editors at LSA. While we understand that achieving this might be somewhat challenging, we do think that having that piece of data is essential, and without it we will not be able to pursue the manuscript further at LSA.

Along with this above-mentioned point, we also encourage you to attend to all the other reviewers' points: re-write the manuscript and revise the data presentation to improve clarity and readers' understanding, either substantiate the development of postsynaptic lattice during synapse formation with additional data or omit this aspect or significantly tone it down, discuss the issues of excitatory vs inhibitory synapses, and discuss the presence of presynaptic and glial components that might indicate difficulty in discriminating between integral components of the lattice vs elements that associate tightly during biochemical purification procedure.

The typical timeframe for revisions is three months. Please note that papers are generally considered through only one revision cycle, so strong support from the referees on the revised

version is needed for acceptance.

Thank you for this interesting contribution to Life Science Alliance. We are looking forward to receiving your revised manuscript.

Sincerely,

Shachi Bhatt, Ph.D.
Executive Editor
Life Science Alliance
<https://www.lsa-journal.org/>
Tweet @SciBhatt @LSAJournal

- A letter addressing the reviewers' comments point by point.
- An editable version of the final text (.DOC or .DOCX) is needed for copyediting (no PDFs).
- High-resolution figure, supplementary figure and video files uploaded as individual files: See our detailed guidelines for preparing your production-ready images, <https://www.life-science-alliance.org/authors>
- Summary blurb (enter in submission system): A short text summarizing in a single sentence the study (max. 200 characters including spaces). This text is used in conjunction with the titles of papers, hence should be informative and complementary to the title and running title. It should describe the context and significance of the findings for a general readership; it should be written in the present tense and refer to the work in the third person. Author names should not be mentioned.

B. MANUSCRIPT ORGANIZATION AND FORMATTING:

Reviewer #1 (Comments to the Authors (Required)):

In this article, Suzuki et al. demonstrate an improved method for the isolation of PSD lattices (i.e. the core cytoskeletal component of post-synaptic densities). PSD isolation and imaging by electron microscopy is an established technique, but little is known about the structural components that allow the formation and stabilization of the PSD protein complexes. One possibility, highlighted in this article, is that a PSD lattice (composed of various cytoskeletal proteins) is the first stable structure to be formed at "proto-synapses", and serves as a backbone for the formation of the PSD per se, including its core components (e.g. PSD95, Homers, Shanks...). A previous study by the same group has previously shown a first method for the isolation of PSD lattices, largely based on sucrose gradient ultracentrifugation of synaptosomal extracts. Unfortunately, the proteins in this preparation could not be analyzed by common Western Blot techniques, presumably due to a first ultracentrifugation step that rendered a lot of the components of the lattice insoluble. To solve this problem, the authors have switched the order of the 2 ultracentrifugation steps (i.e. in the presence and absence of a sucrose gradient), and show that fraction 11 of this new preparation contains PSD lattices, readily observable by electron microscopy (and structurally comparable to previously described PSD electron micrographs). Importantly, the proteins in this fraction are much more soluble and can be run on a WB.

By varying the concentration of the OG detergent used to purify the PSD lattice, this new preparation yields PSD components that bind more or less strongly to the PSD structure: with increasing OG concentration fewer protein interactions are conserved, resulting in sparser structures with a more clear meshwork appearance in EM, termed a "lean" lattice by the authors. Tubulin concentration is similar across these different OG concentrations, but other proteins such as Homer1 or PSD-95 show a decreasing concentration with increasing OG concentration. By using proteomics comparison of these "enriched" and "lean" lattice preparations, the authors then identify a 'minimum essential cytoskeleton or MEC' group of proteins that make up the lean lattice, and a group of non-MEC proteins that contribute to the enriched lattice. This last group of proteins is more loosely associated with the lattice and can be dissociated with increasing detergent concentration.

Thus, by combining proteomic analyses by WB and Mass Spectrometry, as well as previous results from their earlier article, the authors identify the proteins that are part of lean and enriched lattices, and ultimately what they describe as the minimal cytoskeleton components of the PSD lattice. They propose a model in which PSD formation first involves the formation of a non-microtubule tubulin-based backbone. Immunogold EM does indeed show that the purified PSD lattices contain several tubulin isoforms, and also confirms the PSD lattice localization of other identified proteins.

Overall, this study presents interesting observations, but is ultimately a small increment over the previously published results. The manuscript is exceptionally difficult to read, and can only be understood after reading the Suzuki et al. J Neurochem 2018 paper, on which the current manuscript builds. The new PSD lattice preparation appears to be an improvement over the preparation described in that study and could prove useful for neurobiologists' interest in the early steps of synapse formation or in molecular synaptic biology in general. However, the conclusions of the manuscript, in particular the model of an in vivo non-microtubule tubulin-based PSD lattice, are not supported by sufficient experimental evidence. Furthermore, the conclusions based on the new preparation also contradict the previous study (Suzuki et al. 2018), in which it was concluded that

actin, and not tubulin, was a major component of the lattice. This raises doubts on whether biochemical methods can reliably isolate PSD lattice structures.

Major points

- The paper is a very difficult read, mainly because of the nomenclature for the different biochemical preparations, which is extremely confusing. Naming the "old" fraction 1%OG-IS-11U and the "new" fraction 1%OG-11U-IS does not help the reader at all for instance. Moreover, terms with which the authors are no doubt familiar (but with which the reader might not be) need to be described/defined when they appear in the text. Nobody knows what "lean" and "enriched" PSD lattices are, and yet they are used in the abstract. Not everybody knows what the abbreviations SDG, SPM, IAA, OG etc mean. The different rationales for using certain compounds at different concentration need to be made clearer (I still do not know what the properties of OG are that makes it ideal for PSD lattice purification). Overall, this article would greatly benefit from a better readability.
- The authors claim that their PSD lattice structures are representative of an in vivo PSD lattice, but no data is provided to substantiate this point. This is crucial for publication. I would have liked to see EM images (from slices or primary cultures) showing the localization of different tubulin isoforms at the synapse for instance.
- The model that tubulin is able to form non-MT based scaffolds is certainly interesting, and would be very valuable in our thinking of how PSDs develop over time (as the authors suggest in their model). The experimental evidence supporting this model is largely non-existent however. Does the described biochemical technique preserve the integrity of MTs? Does another fraction contain preserved MTs observable by EM? Otherwise, it is a possibility that the presence of monomeric tubulin is due to MT destabilization. Demonstration of non-microtubule tubulin making up the backbone of PSDs by an independent method is required to support this point.
- The authors do not discuss the presence of the presynaptic proteins bassoon or synapsin, or of the glial marker GFAP in the "lean" PSD lattice preparation, which supposedly contains the minimal PSD lattice. It is difficult to imagine how presynaptic or glial proteins could be part of a PSD backbone structure. These proteins are therefore likely to be 'contaminants' that end up in the supposed lattice fraction after centrifugation. This raises doubts on whether a lattice can effectively be isolated from PSD fractions.

Minor points

- The findings in the current manuscript contrast with those in Suzuki et al. 2018, in which it is concluded that actin is the major lattice component, whereas now it is concluded that actin is not a major component (page 5). The discrepancy between the two studies needs to be better discussed.
- The section in which the authors identify MEC non-MEC components is extremely difficult to read and needs to be reworked.
- There are a lot of typos throughout the manuscript and I do not have the time to list them all. The manuscript would greatly benefit from additional proofreading.
- There are missing axis legends which need to be added (ex: Figure 3d).
- The quality of the text in the figures, as well as certain colors, is really bad, likely due to compression artefacts. This is often the case when images are exported in .jpeg and not .png or .tiff.

Reviewer #2 (Comments to the Authors (Required)):

Suzuki and colleagues report on the isolation and characterization of a synaptic protein structure that they call the postsynaptic density lattice. They extend a previous study published in 2018 by

modifying the purification strategy and succeed to perform proteomics analysis on the purified structure, which they claim to be the tubulin-based backbone of the postsynaptic specialization known as postsynaptic density (PSD). They postulate that tubulins are organized in a non-microtubules-like form. In detail, they show electron microscopic images of the lattice structures purified using the detergent octyl glucoside as compared to previous PSD preparations; characterization of these structure by SDS-PAGE and immunoblot analyses; comparison of different lattice fractions using increasing amounts of detergent; comparison of the proteomes of the different fractions using mass spectrometry; and immunogold localization of various protein components including tubulins on the different lattice fractions which they call 'enriched' and 'lean'. They compare postsynaptic lattice preparations from 7 days- and 6 weeks-old rat brains and conclude on the development of these structures. Finally, they sketch a model for the development of the PSD lattice.

Overall the data is interesting for the field and, if the core hypothesis of the existence of non-microtubule tubulin-based structure underneath the postsynaptic membrane can be confirmed, this is a finding that can make it to the textbooks. However, the study, on the one hand, the presentation of the data in manuscript, on the other hand, require further attention.

1) The study does not convincingly show that the tubulin-based lattice exists in brain tissue. This would require, for example, immunogold localization of tubulins within the PSD underneath the postsynaptic membrane, where the authors localize their structure based mainly on biochemical evidence. Alternatively, high-resolution light microscopy may help to collect evidence for such a localization. The authors discuss some, to date sparse, evidence for the localization of tubulin in synaptic dense projection, also the biochemical evidence exists since early days of synaptic protein chemistry in the 1980ies. However, the problem of copurification due to similar biochemical properties or true integral component was not yet solved. The problem is also illustrated by the copurification of mitochondrial proteins in the present preparation, which the authors discuss (most likely correctly) as co-purifying rather integral lattice components.

Even if the localization is solved, the organization of tubulin within the lattice will need to be solved, but this may be far beyond the scope of this study. What is the model for this structure? Is the evidence from the EM data for tubulin dimers?

2) The statements on the development of the PSD-lattice are not justified by the data. In additions to the criticism formulated above more developmental time points need to be analyzed. However, I see that potentially a more detailed analysis during the period of synaptogenesis, when the PSD is formed, may provide an easier access to the in-situ analysis of the tubulin-based subsynaptic lattice.

3) The question appears what type PSD are the authors looking at? In their enriched fraction they find elements of both excitatory and inhibitory synapses. Gephyrin, the major scaffolding protein of inhibitory synapses, is reportedly a tubulin binding protein. Did the authors check for the presence of inhibitory synaptic components in their immunogold studies. It would be a very exciting result, if both types of synapse had a very similar basic lattice even though the appearances in situ look quite different.

4) The presentation of data in the manuscript is quite complex and even experts familiar with the field have are hard time to work through it. This also applies to the discussion. A clearer and more reflective presentation, documenting that this is an interesting step forward on the way to solve the question whether there is non-microtubule tubulin-based backbone within the core of synapses would be adequate. The statement on the developmental role at this point is pure speculation and should not appear in the abstract.

Specific points:

Page 6: "Protein recovery in the PSD lattice preparations increased with the detergent concentration (Fig. 3b)." What is meant with this statement? Why is the recovery higher with higher detergent concentrations? I assume the recovery from the supernatant is meant. Right? Please, clarify.

Page 11: the authors state: "After purifying a novel PSD lattice preparation, we acquired novel information on its components and established a new PSD lattice model consisting of a non-MT tubulin-based backbone and its associated proteins".

I feel, the word "novel" is not really appropriate, actually in both cases.

Fig. 1: The comparison of 7d and 6w lattices is very superficial (see also page 4). As a minimum a quantitative assessment would be expected. Clearly, from two time points a time course of formation cannot be deduced (cf. major point 2).

In Fig. 3d Homer 1 is basically absent in Western blots (fig. 3d) but present in EM and MS analyses (Fig 5B,C, Suppl tables 1 and 2). Is there any explanation for this?

Fig. 4 / Table S3: Maybe it would be interesting to compare the MS data on the different proteomes with the SynGO (synaptic gene ontology) database to extract further information (Koopmans et al., 2019). This might help to define 'true' synaptic proteins.

Discussion (page 12):

The presence of tubulin in spines does not necessarily mean that it is in the PSD (cf. major point 1)

Abbreviations should be defined when they first appear, and not only in the abbreviation list, to increase readability. Some may not even be necessary (e.g. SDG or a-IN). Some are missing (e.g. # AA).

References need editing. Some are incomplete (e.g. Hu et al., 2002) and journal abbreviations are inconsistent.

Fig. 5C: What is the difference between Ca and Cb? Is 5Cb wrongly assigned?

Fig S2: Replace "pelett" by "pellet".

Reviewer #3 (Comments to the Authors (Required)):

1. In the current manuscript, Suzuki et al. described newly developed improved method to biochemically isolate a structural backbone of postsynaptic density (PSD) in excitatory synapses, called "PSD lattices".

Using this new method, they have now identified a major component of the PSD lattice is a non-microtubule tubulin together with other protein components, which was impossible to be determined by previous methods due to insolubility of the protein components in the lattice. The lattices the authors could biochemically isolate might provide structural bases for a trans-synaptic nanocolumn and/or the "slots" for postsynaptic proteins and their complexes. The

biochemical work together with electron microscopy which authors have performed will compliment recent advance in the studies of synapses using super resolution microscopy to elucidate organisational principles of postsynaptic density.

Detailed investigation on the organisational principals of the postsynaptic density structures and their dynamics will provide a key to understand how complex information processing is achieved in the excitatory synapses. Therefore, the work presented by Suzuki et al. in the current manuscript will be very valuable to the wide range of neuroscience and biochemical communities, and I recommend publishing this work in Life Science Alliance.

2. The authors developed a new method to purify the PSD lattices and compared these with the sample purified by the previous methods by using biochemical analysis and electron microscopy. Data presented were supportive to show the advantage of their new method.

The authors analysed the protein components in the new PSD lattices and compared with that of the old one in which actin was the main components. The authors could identify many other proteins in the new lattice preparation, these including the α and β isoforms of tubulin, ATP synthase α , β , and ATP/ADP translocase as well as other signalling and structural molecules. They could see two structurally distinct types of the lattices (enriched-type PSD lattice and lean-type PSD lattice) by EM, depending on the purification conditions.

Data seemed strongly supportive for their new findings.

The authors further compared the component of the PSD lattices purified by 3 different preparations (initial PSD lattice by their previous method, enriched-type PSD lattice and lean-type PSD lattice) by shot-gun mass-spec. They categorised these proteins into the minimum essential cytoskeleton (MEC) proteins and non-MEC proteins depending on these in the different biochemical fractions. The non-MEC proteins seemed to be the ones that could be structurally organised by the core lattice formed by the MEC proteins. Suzuki's group is one of the pioneers to perform biochemical analysis of PSD proteins and these proteomic data seemed solid.

The authors have investigated the localization of the tubulin and other main so called major PSD molecules, such as CaMKIIs, PSD-95, Shank, GKAP and Homer, in the PSD lattice by using immunogold negative staining EM. This further convinced that these molecules were associated with the PSD lattice, may be in the different organisational principles. Data looks solid but this may need more investigation in future.

3. It would help if the authors can briefly describe what the different biochemical fractions (e.g. "1%OG-11U-IS" etc.) mean in the early part of the main text.

It would be interesting if authors can mention about the relationship between the relative abundance of each protein in PSD and the protein found in their 2 categorisations; MEC and non-MEC proteins.

Thank you for the review and valuable comments. We revised the manuscript considering the reviewer's comments and advice. We believe that we have cleared all problems specified by the reviewers by this revision.

1. (Reviewer) Overall, this study presents interesting observations, but is ultimately a small increment over the previously published results.

-> As pointed out, this manuscript is an extended version of our previous publication, and the model we proposed is not so much changed. However, the previous preparation was not good enough because of possible contamination of non-synaptic proteins by inappropriate aggregation. In this manuscript, we could identify tubulin as a key component and we obtained supporting data that demonstrates tubulin immunoreactivity in the *in situ* PSD in the new lattice preparation. This manuscript provides clear difference between PSD lattice backbone structure and the assembly of PSD scaffold and adaptor proteins. Thus, we believe the additional points to the new manuscript are not small.

2. (Reviewer) The manuscript is exceptionally difficult to read,

-> Sorry for the complexity in the initial manuscript. English and presentation have been extensively revised in this revision. A native English speaker in Editage proofreading company proofread the manuscript.

3. (Reviewer) the conclusions of the manuscript, in particular the model of an *in vivo* non-microtubule tubulin-based PSD lattice, are not supported by sufficient experimental evidence.

-> We carried out an additional experiment upon strong request by the Editor to provide supporting data for the non-microtubule tubulin-based PSD lattice model. We chose immunohistochemistry at EM level using immuno-gold labeling method, one of the approach suggested by the Editor. In the additional experiment, we could demonstrate the presence of tubulin-immunoreactivity in the PSD and its neighbors. This tubulin may be in non-microtubule form in the brain because microtubules are rarely seen in the PSD (by our data and many other publications). Furthermore, we added the discussion that the tubulin-based lattice structures are not artificial structure formed after detergent solubilization of SPM. Therefore, we think sound supporting data were added to the revised manuscript.

4. (Reviewer) the conclusions based on the new preparation also contradict the previous study (Suzuki et al. 2018), in which it was concluded that actin, and not tubulin, was a major component of the lattice. This raises doubts on whether biochemical methods can reliably isolate PSD lattice structures.

-> The initial PSD lattice and new PSD lattice have some differences in their components such as actin. This is due to the fact that proteins that are not directly linked to *in situ* PSD lattices are more abundant in the initial PSD lattice than the new PSD lattice (Fig. 4a, 4b). The initial PSD lattice preparation was not a good one because of relatively large amount of incorporation of non-synaptic proteins by inappropriate aggregation during purification. Thus, initial and the new PSD lattice preparations demonstrate a difference in their components (the latter is more physiological and less contaminated). Therefore, this is not a discrepancy. These things have been written in the text* and following sentence was further added to the revised manuscript to avoid the misunderstanding by readers: "The higher content of actin and α -IN in the initial PSD lattices than in the new PSD lattices may be because proteins, such as actin and α -IN, which are not closely linked to *in situ* PSD lattices, are abundant in the initial PSD lattice preparation (Fig. 4a, 4b)." (in the last part of the section "Tubulin is a major minimum essential cytoskeleton (MEC) protein").

*Following is related descriptions in the initial manuscript and in the revised manuscript.

"Thus, an additional ultracentrifugation step before the SDG ultracentrifugation step appeared to make some proteins insoluble in SDS, most likely due to the fact that highly concentrated conditions of a large variety of proteins around the PSD lattice caused extensive protein-protein interactions, resulting in protein aggregation (Zeng et al., 2016)."

"1%OG-IS-11B (initial PSD lattice) contained a relatively large number of proteins that were absent in the 11U-IS samples (new PSD lattices). A large number of proteins may be artificially associated

with the PSD lattice structure under forced concentrated conditions by ultracentrifugation in the initial purification protocol.”

“The robust concentration of the PSD lattice structure after ultracentrifugation in the presence of a large number of non-neighboring proteins immediately after detergent treatment of the SPM appears to induce their association with the PSD lattice structure.”

Major points

5. (Reviewer) The paper is a very difficult read, mainly because of the nomenclature for the different biochemical preparations, which is extremely confusing. Naming the "old" fraction 1%OG-IS-11U and the "new" fraction 1%OG-11U-IS does not help the reader at all for instance. Moreover, terms with which the authors are no doubt familiar (but with which the reader might not be) need to be described/defined when they appear in the text. Nobody knows what "lean" and "enriched" PSD lattices are, and yet they are used in the abstract. Not everybody knows what the abbreviations SDG, SPM, IAA, OG etc. mean. The different rationales for using certain compounds at different concentration need to be made clearer (I still do not know what the properties of OG are that makes it ideal for PSD lattice purification). Overall, this article would greatly benefit from a better readability.

-> We are very sorry for the defects in readability of the initial manuscript. Now we improved the style of the manuscript in regard to abbreviations, nomenclature of the preparations and complexity of their usage. We changed “lean and enriched PSD lattice” to “lean- and enriched-type PSD lattice in the abstract because it is difficult and inappropriate to state the details in an abstract. Please understand. Next, conditions of experiments using OG are mostly empirical ones derived from our previous studies on the postsynaptic rafts**, and did not based on limited information on the properties of the detergent in the literature. We have cited our previous publications.

** Suzuki, T., J. Zhang, S. Miyazawa, Q. Liu, M.R. Farzan, and W.D. Yao. 2011. Association of membrane rafts and postsynaptic density: proteomics, biochemical, and ultrastructural analyses. *Journal of Neurochemistry*. 119:64-77. (This paper is not cited in the manuscript.)

Liu, Q., W.-D. Yao, and T. Suzuki. 2013. Specific interaction of postsynaptic densities with membrane rafts isolated from synaptic plasma membranes. *Journal of Neurogenetics*. 27:43-58.

Zhao, L., H. Sakagami, and T. Suzuki. 2014. Detergent-dependent separation of postsynaptic density, membrane rafts and other subsynaptic structures from the synaptic plasma membrane of rat forebrain. *Journal of Neurochemistry*. 131:147-162.

6. (Reviewer) The authors claim that their PSD lattice structures are representative of an in vivo PSD lattice, but no data is provided to substantiate this point. This is crucial for publication. I would have liked to see EM images (from slices or primary cultures) showing the localization of different tubulin isoforms at the synapse for instance.

-> Please read the No. 3 response (above).

7. (Reviewer) The model that tubulin is able to form non-MT based scaffolds is certainly interesting, and would be very valuable in our thinking of how PSDs develop over time (as the authors suggest in their model). The experimental evidence supporting this model is largely non-existent. Does the described biochemical technique preserve the integrity of MTs? Does another fraction contain preserved MTs observable by EM? Otherwise, it is a possibility that the presence of monomeric tubulin is due to MT destabilization. Demonstration of non-microtubule tubulin making up the backbone of PSDs by an independent method is required to support this point.

-> There is no report that tells about MT polymerization-depolymerization conditions during biochemically purifying PSD or PSD-related structures. Our “non-MT tubulin model” was based on the two findings: 1) Tubulin on the purified PSD lattice is in a non-MT form because there are no MT in the

purified PSD lattice. 2) Tubulin-immunoreactivity is present in the *in situ* PSD, where no MT was present at EM level under the conditions maximally preserving MT (Westrum et al., 1980*). Therefore, it is highly plausible that tubulin in the *in situ* PSD lattice and PSD is in a non-MT form. We proposed non-microtubule-based backbone structure as a model worth verifying. We stated this point in the chapter "Tubulin is a major MEC protein."

*Westrum, L.E., D.H. Jones, E.G. Gray, and J. Barron. 1980. Microtubules, dendritic spines and spine apparatuses. *Cell and tissue research*. 208:171-181.

8. (Reviewer) The authors do not discuss the presence of the presynaptic proteins bassoon or synapsin, or of the glial marker GFAP in the "lean" PSD lattice preparation, which supposedly contains the minimal PSD lattice. It is difficult to imagine how presynaptic or glial proteins could be part of a PSD backbone structure. These proteins are therefore likely to be 'contaminants' that end up in the supposed lattice fraction after centrifugation. This raises doubts on whether a lattice can effectively be isolated from PSD fractions.

-> Contamination or purity problem of the subcellular purified material is a well-known problem for a long time. Whether integral proteins or contamination, is an unavoidable problem to biochemically-purified structures. In fact, the same problem is also observed in the established PSD preparation: glial components, presynaptic elements, inhibitory synapse components, nuclear proteins, and mitochondrial proteins are present in the purified PSD. The only way to solve this problem is one-by-one localization analysis of proteins by immunohistochemical or immunocytochemical approach. We avoided the discussion on this matter in the first manuscript because others and we have commented on this before, did not focus on this matter in the present paper, and have not tested these proteins intensively. In the preliminary immuno-gold negative EM using anti-gephyrin antibody, which was done after the submission of the 1st manuscript, did not detect immunoreactive signals on the PSD lattice. In the revised manuscript we added brief comments on this matter in response to the reviewer's question (last part in the discussion "Non-MEC proteins and...").

Minor points

9. (Reviewer) The findings in the current manuscript contrast with those in Suzuki et al. 2018, in which it is concluded that actin is the major lattice component, whereas now it is concluded that actin is not a major component (page 5). The discrepancy between the two studies needs to be better discussed.

-> Please read above (No. 4).

10. (Reviewer) The section in which the authors identify MEC non-MEC components is extremely difficult to read and needs to be reworked.

-> I am very sorry for the complexity. I agree with the reviewer. The portion was meaninglessly too precise. We have made it concise. In particular, the 3rd chapter (Proteins in the non-MEC group) was extensively revised. Fig. 4 was also changed for easy understanding by replacing the exact preparation name.

11. (Reviewer) There are a lot of typos throughout the manuscript and I do not have the time to list them all. The manuscript would greatly benefit from additional proofreading.

-> I am very sorry. They were thoroughly checked.

12. (Reviewer) There are missing axis legends which need to be added (ex: Figure 3d).

-> We added the "protein amount (%)" and "(amount) of the measured protein" in the figure and its legend, respectively.

13. (Reviewer) The quality of the text in the figures, as well as certain colors, is really bad, likely due to compression artifacts. This is often the case when images are exported in .jpeg and not .png or .tiff.

-> I am very sorry for the poor quality. I amended them.

MS data from this publication have been deposited to jPOSTrepo (an international standard data repository for proteomes, <https://repository.jpostdb.org>)*. The accession numbers are PXD024712 for ProteomeXchange (<http://www.proteomexchange.org>) and JPST001106 for jPOST (Japan Proteome Standard DataBase, <https://jpostdb.org>). For confidential access, please use User name (suzukit@shinshu-u.ac.jp), Passwords (jPostPSDLaticerepo) at <https://repository.jpostdb.org/mypage>.

To the Reviewer #2:

Thank you for the review and valuable comments. We revised the manuscript considering reviewer's comments and advice. We greatly appreciate your positive evaluation to the essential part on this study, but very sorry for the defects in the readability of the initial manuscript. We believe that we have cleared all problems specified by the reviewers by this revision

1-1) The study does not convincingly show that the tubulin-based lattice exists in brain tissue. This would require, for example, immunogold localization of tubulins within the PSD underneath the postsynaptic membrane, where the authors localize their structure based mainly on biochemical evidence. Alternatively, high-resolution light microscopy may help to collect evidence for such a localization.

-> We carried out an additional experiment upon strong request by the Editor and editors to provide supporting data for the non-microtubule tubulin-based PSD lattice model. We chose immunohistochemistry at EM level using immuno-gold labeling method, one of the approaches suggested by the Editor and reviewers. In the additional experiment, we could demonstrate the presence of tubulin-immunoreactivity in the PSD (both PSD core including near the membrane region, and pallium). This tubulin may be in non-microtubule form in the brain because microtubules are rarely seen in the PSD (by our data and many other publications). Furthermore, we added the discussion that the tubulin-based lattice structures are not an artificial structure formed after detergent solubilization of SPM. The discussion section "Tubulin is a major minimum essential cytoskeleton (MEC) protein" was extensively reorganized with the results of the additional experiment.

1-2) The authors discuss some, to date sparse, evidence for the localization of tubulin in synaptic dense projection, also the biochemical evidence exists since early days of synaptic protein chemistry in the 1980ies. However, the problem of copurification due to similar biochemical properties or true integral component was not yet solved....

-> Contamination or purity problem of the subcellular purified material is a well-known problem known for a long time. A question whether integral proteins or contamination, is an unavoidable problem to biochemically purified structures. In fact, the same problem is also observed in the established PSD preparation: glial components, presynaptic elements, inhibitory synapse components, nuclear proteins, and mitochondrial proteins are present in the purified PSD. The only way to solve this problem is one-by-one localization analysis of proteins by immunohistochemical or immunocytochemical approach. We avoided the discussion on this matter in the first manuscript because others and we have commented on this before, did not focus on this matter in the present paper, and have not tested these proteins intensively. In the preliminary immuno-gold negative EM using anti-gephyrin antibody, which was done after the submission of the 1st manuscript, did not detect immunoreactive signals on the PSD lattice. In the revised manuscript we added brief comments on this matter in response to the reviewer's request (last part in the discussion "Non-MEC proteins and...."). Despite this unavoidable problem to subcellular fractionation, contamination to the purified PSD lattice was reduced by the improvement of the purification protocol (the PSD lattice prepared by the new method is less contaminated than those purified by the initial method). Also, we commented that purified PSD lattice might exist as integral structure (The discussion section "Tubulin is a major minimum essential cytoskeleton (MEC) protein").

1-3) Even if the localization is solved, the organization of tubulin within the lattice will need to be solved, but this may be far beyond the scope of this study. What is the model for this structure? Is the evidence from the EM data for tubulin dimers?

-> There is no model for the proposed lattice tubulin-based structure of the PSD lattice. We don't know whether lattice tubulin is in a dimer form. We just stated as "non-MT form". Our "non-MT tubulin model" was based on the two findings: 1) Tubulin on the purified PSD lattice is in a non-MT form because there are no MT in the purified PSD lattice. 2) Tubulin-immunoreactivity is present in the *in situ* PSD, where no MT was present at EM level under the conditions maximally preserving MT (Westrum et al., 1980*). Therefore, it is highly plausible that tubulin in the *in situ* PSD lattice and PSD is in a non-MT form. We proposed non-MT-based backbone structure as a model worth verifying. We stated this point in the chapter "Tubulin is a major minimum essential cytoskeleton (MEC) protein."

*Westrum, L.E., D.H. Jones, E.G. Gray, and J. Barron. 1980. Microtubules, dendritic spines and spine apparatuses. *Cell and tissue research*. 208:171-181.

2) The statements on the development of the PSD-lattice are not justified by the data. In addition to the criticism formulated above more developmental time points need to be analyzed.

-> We toned down our developmental model on the PSD through modification of PSD lattice. We changed the previous developmental model covering entire developmental stages of PSD (previous Fig. 6) to the concept focusing on the relationship between the PSD lattice backbone structure and lean-/enriched- type PSD lattices (new Fig. 8).

3) The question appears what type PSD are the authors looking at? In their enriched fraction they find elements of both excitatory and inhibitory synapses. Gephyrin, the major scaffolding protein of inhibitory synapses, is reportedly a tubulin binding protein. Did the authors check for the presence of inhibitory synaptic components in their immunogold studies? It would be a very exciting result, if both types of synapse had a very similar basic lattice even though the appearances *in situ* look quite different.

-> Overall morphology of the PSD lattice we studied resembles type I excitatory PSD. Thus, we think PSD lattices we purified are mostly related to type I PSDs. In fact, proteomics data identified a number of components of type I excitatory PSD. However, it also contained components of inhibitory synapse, such as gephyrin and GABA receptors. We did immuno-gold negative EM using anti-gephyrin antibody after the submission of the 1st manuscript, but did not detect immunoreactive signals on the PSD lattice. We briefly commented about contamination problems accompanied to the subcellular fractionation in the text. See also above (1-2).

4) The presentation of data in the manuscript is quite complex and even experts familiar with the field have a hard time to work through it. This also applies to the discussion. A clearer and more reflective presentation, documenting that this is an interesting step forward on the way to solve the question whether there is non-microtubule tubulin-based backbone within the core of synapses would be adequate. The statement on the developmental role at this point is pure speculation and should not appear in the abstract.

-> I am very sorry for the complexity of the initial manuscript. We revised it extensively. We removed "developmental" in the last sentence in the abstract in accordance with the reviewer's point out.

Specific points:

5) Page 6: "Protein recovery in the PSD lattice preparations increased with the detergent concentration (Fig. 3b)." What is meant with this statement? Why is the recovery higher with higher detergent concentrations? I assume the recovery from the supernatant is meant. Right? Please, clarify.

-> I changed "Protein recovery in the PSD lattice preparations" to "Protein recovery of the PSD lattice preparations". The result is that the recovery of the PSD lattice increased in parallel to the increase of detergent concentration. That is, the extraction of the PSD lattice structure was increased when the detergent concentration was increased from 0.75% to 5%.

6) Page 11: the authors state: "After purifying a novel PSD lattice preparation, we acquired novel information on its components and established a new PSD lattice model consisting of a non-MT

tubulin-based backbone and its associated proteins". I feel, the word "novel" is not really appropriate, actually in both cases.

-> We changed to "After purifying a new PSD lattice preparation, we acquired information on its components and established a new PSD lattice model consisting of..."

7) Fig. 1: The comparison of 7d and 6w lattices is very superficial (see also page 4). As a minimum a quantitative assessment would be expected. Clearly, from two time points a time course of formation cannot be deduced (cf. major point 2).

-> We changed the previous developmental model covering entire developmental stages of PSD (Fig. 6) to the concept focusing on the relationship between the PSD lattice backbone structure and lean-/enriched- type PSD lattices (new Fig. 8).

8) In Fig. 3d Homer 1 is basically absent in Western blots (fig. 3d) but present in EM and MS analyses (Fig 5B,C, Suppl tables 1 and 2). Is there any explanation for this?

-> This sometimes happens due to the difference in the sensitivity in the MS and Western blotting. Sensitivity of MS is very high. Therefore, MS sometimes detects proteins that are not detected by WB. Immuno-gold EM can identify a single molecule, if present. Thus, it produces a difference from the result of WB. In the text, we carefully chose "under detection level" in the detection by WB. We guess WB will not detect them even if amounts are increased to, for example, two-fold.

9) Fig. 4 / Table S3; Maybe it would be interesting to compare the MS data on the different proteomes with the SynGO (synaptic gene ontology) database to extract further information (Koopmans et al., 2019). This might help to define 'true' synaptic proteins.

-> Thank you for the suggestion. Comparison of our lattice data with PSD proteomes is interesting work and may be helpful to our PSD lattice research project. However, addition of proteome comparison to this paper is beyond major concern in this paper and it may be difficult to add data to the already huge manuscript. We would like to try it in another opportunity.

10) Discussion (page 12):

The presence of tubulin in spines does not necessarily mean that it is in the PSD (cf. major point 1)

-> "Spine" was changed to "spine head," where PSDs are located.

11) Abbreviations should be defined when they first appear, and not only in the abbreviation list, to increase readability. Some may not even be necessary (e.g. SDG or a-IN). Some are missing (e.g. # AA).

-> Sorry for the inconvenience. In this revision, the manuscript was thoroughly checked and this problem was amended. We also changed "#" to "No. of" and replaced "AA" by "amino acids" in all Tables and Supplementary Tables. We also added comment on the naming of the subcellular fractions to the site of the first appearance.

12) References need editing. We overlooked this error. Some are incomplete (e.g. Hu et al., 2002) and journal abbreviations are inconsistent.

-> This was EndNote error. We overlooked this error. We corrected the reference information of Hu et al. Sorry for the defect in the critical reference.

13) Fig. 5C: What is the difference between Ca and Cb? Is 5Cb wrongly assigned?

-> Fig 5Ca and Cb show different type structures present in the same sample (0.75% OG). This has been stated in the legend. We showed this structure because we excluded this type of structure for quantitation assay. "(Cb)" was changed to "(C-b)" (the Fig. 5 legend).

14) Fig S2: Replace "pelett" with "pellet".

-> We corrected this overlooked mistake. Sorry for this error.

MS data from this publication have been deposited to jPOSTrepo (an international standard data repository for proteomes, <https://repository.jpostdb.org>) *. The accession numbers are PXD024712 for ProteomeXchange (<http://www.proteomexchange.org>) and JPST001106 for jPOST (Japan Proteome Standard DataBase, <https://jpostdb.org>). For confidential access, please use User name (suzukit@shinshu-u.ac.jp), Passwords (jPostPSDlaticerepo) at <https://repository.jpostdb.org/mypage> .

To the Reviewer #3:

Thank you for the evaluation of our work. We added further information supporting our non-MT tubulin-based lattice model by additional experimental data. We also reorganized the manuscript extensively for the readers to understand easily. We believe that the quality of the paper was improved extensively and the paper became more persuasive by this revision.

MS data from this publication have been deposited to jPOSTrepo (an international standard data repository for proteomes, <https://repository.jpostdb.org>) *. The accession numbers are PXD024712 for ProteomeXchange (<http://www.proteomexchange.org>) and JPST001106 for jPOST (Japan Proteome Standard DataBase, <https://jpostdb.org>). For confidential access, please use User name (suzukit@shinshu-u.ac.jp), Passwords (jPostPSDlaticerepo) at <https://repository.jpostdb.org/mypage> .

April 9, 2021

RE: Life Science Alliance Manuscript #LSA-2020-00945-TR

Prof. Tatsuo Suzuki
Shinshu University
Molecular and Cellular Physiology
3-1-1, Asahi
Matsumoto 390-8621
Japan

Dear Dr. Suzuki,

Thank you for submitting your revised manuscript entitled "Non-microtubule tubulin-based backbone and subordinate components of postsynaptic density lattices". Your manuscript has been re-reviewed by the referees and further assessed by our team of editors. The referee reports are appended at the end of this email.

While we understand Reviewer 1's concern that whether the biochemical preparation included in the manuscript truly represents a lattice PSD and which proteins are true constituents of such a lattice remains difficult to judge, we also understand that it might be difficult for you to address these issues experimentally at this stage. Instead, we would ask you to add a clear statement in the manuscript that the possibility that preparation artefacts may contribute to the observed pattern cannot be fully ruled out.

Once these requested edits are made, the presentation concerns raised by Reviewer 2 are addressed, and below-mentioned formatting changes are included, the revised manuscript should be ready to be published in Life Science Alliance.

Please address the following in the revision (along with additional points mentioned below):

- please upload your main and supplementary figures as single files
- please make sure the author order in your manuscript and our system match
- please use the [10 author names, et al.] format in your references (i.e. limit the author names to the first 10)
- please use Capital letter when introducing the panels in the legend for Figures 3, 4, 6, S1, S2, in actual Figures and their callouts in the manuscript text
- please revise the legend for figure 5 so that the panels are introduced in order
- please upload your main manuscript text as an editable doc file;
- please upload your Tables in editable .doc or excel format
- please add callouts for Figures 1Bb, Bd, 5Ca-l, 6a,d,e,r to your main manuscript text
- please provide higher resolution images for all the blots included in Figure 2, Figure 3 and Figure 6R

To upload the final version of your manuscript, please log in to your account:
<https://lsa.msubmit.net/cgi-bin/main.plex>

A. FINAL FILES:

B. MANUSCRIPT ORGANIZATION AND FORMATTING:

Thank you for your attention to these final processing requirements. Please revise and format the

manuscript and upload materials within 7 days.

Sincerely,

Shachi Bhatt, Ph.D.
Executive Editor
Life Science Alliance
<http://www.lsajournal.org>
Tweet @SciBhatt @LSAJournal

Reviewer #1 (Comments to the Authors (Required)):

The authors have improved the readability of their manuscript. They have also added new experiments that show the presence of tubulin immunoreactivity near PSDs in situ. Their new results support the presence of tubulin near PSDs, but the nature of this tubulin (microtubular or non-microtubular) is not known. The authors' argument that tubulin immunoreactivity in the synaptic cleft may represent extracellular tubulin appears unlikely; a more likely explanation is non-specific labeling. It further remains very difficult to judge whether the presence of tubulin in the PSD-lattice is not an artefact of the biochemical procedure, as the authors argue other proteins present in this preparation (presynaptic, glial etc) are. Why would these be contaminants but tubulin be a bona fide constituent? Finally, the nature of non-microtubule tubulin in the lattice, if it is there, remains unclear. It is suggested throughout the manuscript that the lattice contains tubulin, which suggests a contribution of tubulin to a cytoskeleton-like meshwork (as also drawn in the cartoon in Figure 8). But how can monomeric or dimeric tubulin provide such a structural support if not tubular? Overall I appreciate the authors' efforts to improve their manuscript. They show tubulin is present near PSDs. Whether their biochemical preparation truly represents a lattice PSD, and which proteins are true constituents of such a lattice, remains difficult to judge in my opinion.

Reviewer #2 (Comments to the Authors (Required)):

The revised version of the manuscript by Suzuki and colleagues includes now an immunogold electron microscopic localization analysis for tubulins at excitatory synapses of the murine cerebral cortex (new Fig. 6). Quantitative analysis reveals indeed an enrichment of tubulin immunoreactivity in core region, but also in the pallium region of the postsynaptic density. The authors have also addressed my concerns about the strong statement on the development of the postsynaptic density. They also clarify that they could not identify type II postsynaptic structures in their EM data, although gephyrin is found in the biochemical preparations. Somewhat enigmatic is still the high amount of mitochondrial proteins in the core PSD preparation. However, the authors discuss this intensely and provide some possible explanations. The manuscript still suffers from its substantial complexity, which makes it very difficult to read and comprehend. This is mainly due to the very complicated nomenclature the authors invented for their different fractions that they analyze. Maybe they can find a still better way to illustrate the different fractions. Currently, designations of protein fractions are based on a mixture of positions in

the sucrose density gradient, concentrations of detergent and step in the procedure, and are opposed to terms like MEC vs. non-MEC and enriched-type vs. lean-type, initial vs. new PSD lattice and conventional PSD, and TX-PSD vs OG-PSD. To understand all this one has to toggle between Fig. S1 and Fig. 4. A unified scheme would also help to appreciate the multiple tables and supplementary tables of the manuscript.

April 19th, 2021

Re: Life Science Alliance manuscript #LSA-2020-00945-TRR

Shachi Bhatt, Ph.D.
Executive Editor
Life Science Alliance

Dear Editor

I am very glad to hear from you the review results to our revised manuscript.

1) First of all, we added the following statement according to your instruction.
“Further approaches might be necessary to fully rule out the possibility that preparation artifacts may contribute to the observed pattern, however, it is highly plausible under the present conditions that tubulin is an integral component of the PSD lattice and a key molecule that forms the structure under present conditions.”

2) To address towards concerns of the Reviewer 2, we changed the preparation names, and make Supplementary Fig 1 in the previous version to new Fig. 1, according to the reviewer's advice. To accommodate this change, we transferred previous Suppl. Fig. 1D to new Fig. 1 as Fig. 1A. We also abbreviated PSD lattice to PSDL to simplify the preparation name.

Preparation names must have no space before parenthesis, such as PSDL(1%OG, U, 7d). Otherwise, misunderstanding will happen in some cases. I would be very happy if you accept this style.

3) Minor revision that is not requested but necessary for quality of the paper. These revision will not affect on the conclusion.

Fig. 9 cartoon of the backbone structure: small particles on the left most structure were removed.

Following sentence was removed from the text because of scanty of type II PSDs identified. “There was no immuno-gold inside the type II PSDs.”

Possibility of nonspecific signal for cleft gold particles was added in the text.

Following sentence was added. “It is currently completely unknown how non-MT tubulin plays a structural role in the PSDL.”

I replaced WB images, however, the original raw images are unfortunately too small. So, I am not sure whether the resolution is improved.

We hope that everything that has been requested has been properly addressed.

Sincerely,
Tatsuo SUZUKI

April 26, 2021

RE: Life Science Alliance Manuscript #LSA-2020-00945-TRR

Prof. Tatsuo Suzuki
Shinshu University
Molecular and Cellular Physiology
3-1-1, Asahi
Matsumoto 390-8621
Japan

Dear Dr. Suzuki,

Thank you for submitting your Research Article entitled "Non-microtubule tubulin-based backbone and subordinate components of postsynaptic density lattices". It is a pleasure to let you know that your manuscript is now accepted for publication in Life Science Alliance. Congratulations on this interesting work.

DISTRIBUTION OF MATERIALS:

Again, congratulations on a very nice paper. I hope you found the review process to be constructive and are pleased with how the manuscript was handled editorially. We look forward to future exciting submissions from your lab.

Sincerely,

Shachi Bhatt, Ph.D.

Executive Editor

Life Science Alliance

<http://www.lsajournal.org>
